# Rethinking Adversarial Policies: A Generalized Attack Formulation and Provable Defense in RL

**Xiangyu Liu[1], Souradip Chakraborty[1], Yanchao Sun[2], Furong Huang[1]**
[1]University of Maryland, College Park, [2]J.P. Morgan AI Research
{xyliu999}@umd.edu

## Abstract

Most existing works focus on direct perturbations to the victim's state/action or the underlying transition dynamics to demonstrate the vulnerability of reinforcement learning agents to adversarial attacks. However, such direct manipulations may not be always realizable. In this paper, we consider a multi-agent setting where a well-trained victim agent $\nu$ is exploited by an attacker controlling another agent $\alpha$ with an *adversarial policy*. Previous models do not account for the possibility that the attacker may only have partial control over $\alpha$ or that the attack may produce easily detectable "abnormal" behaviors. Furthermore, there is a lack of provably efficient defenses against these adversarial policies. To address these limitations, we introduce a generalized attack framework that has the flexibility to model to what extent the adversary is able to control the agent, and allows the attacker to regulate the state distribution shift and produce stealthier adversarial policies. Moreover, we offer a provably efficient defense with polynomial convergence to the most robust victim policy through adversarial training with timescale separation. This stands in sharp contrast to supervised learning, where adversarial training typically provides only *empirical* defenses. Using the Robosumo competition experiments, we show that our generalized attack formulation results in much stealthier adversarial policies when maintaining the same winning rate as baselines. Additionally, our adversarial training approach yields stable learning dynamics and less exploitable victim policies.[1]

## 1 Introduction

Despite the huge success of deep reinforcement learning (RL) algorithms across various domains (Silver et al., 2017; Mnih et al., 2015; Schulman et al., 2015), it has been shown that deep reinforcement learning policies are highly vulnerable to adversarial attacks. That is, a well-trained agent can produce wrong decisions under small perturbations, making it risky to deploy RL agents in real-life applications with noise and high stakes. The most popular attack methods focus on fooling the RL agent by adversarially perturbing the states, actions, or transition dynamics of the victim (Huang et al., 2017; Pattanaik et al., 2017; Zhang et al., 2020b; Sun et al., 2021; Tessler et al., 2019).

However, in practice, many applications, such as air traffic control systems, are well-protected, meaning that direct perturbations to the observations or actions may not always be feasible. For instance, to perturb the readings of an air traffic control radar, an attacker might need to physically manipulate the sensor or infiltrate the communication system, tasks that can require significant effort. In this context, our paper explores attacks on a victim agent, $\nu$, executed by an attacker controlling another agent, $\alpha$, in the same environment. Specifically, in the air traffic control example, an attacker could manipulate a commercial drone to interfere with the radar system of the victim. The strategy employed by the attacker in this scenario is known as an "*adversarial policy*".

Previous works (Gleave et al., 2019; Wu et al., 2021b; Guo et al., 2021) have adopted principled approaches to attack well-trained RL agents by developing an adversarial policy to directly minimize

---

[1]Codes are available at https://github.com/xiangyu-liu/Rethinking-Adversarial-Policies-in-RL.git.

the expected return of the victim. These methods have effectively defeated state-of-the-art agents trained through self-play (Bansal et al., 2018), even when the adversarial policy has been trained for less than 3% of the self-play training time steps. However, existing models do not adequately address situations where the attacker might face resistance and only achieve partial control, which also lead to conspicuous behaviors. Despite the common occurrence of such attacks, provably efficient defenses are not well investigated yet.

To address these issues, our generalized attack formulation introduces an "attack budget", effectively capturing the attacker's *partial control*. This metric accurately reflects the attacker's capacity to degrade a victim's performance. Within this framework, the attacker can self-regulate the attack budget, aligning with state distributions (Gleave et al., 2019) and marginalized transition dynamics (Russo & Proutiere, 2021; Franzmeyer et al., 2022) to craft stealthier, less detectable attacks. Notably, our attack model extends single-agent action adversarial RL (Tessler et al., 2019) to multi-agent setting. On the defense side, merely retraining the victim agent against specific strong attacks may not necessarily improve overall robustness; in some cases, it could even worsen performance against other potential attacks. We propose an adversarial training algorithm featuring *timescale separation*, which avoids overfitting to specific attacks and focuses on optimizing the agent's worst-case performance. Unlike existing methods of timescale separation in GANs and adversarial training in supervised learning, which may converge to local solutions (Heusel et al., 2017) or provide only empirical defenses (Madry et al., 2017; Shafahi et al., 2019), our algorithm converges to the most robust policy globally, offering defenses with provably efficient guarantees — even in the face of the problem's non-convexity and non-concavity.

Our key contributions in the realm of both attack and defense are summarized as follows. **(1)** We introduce a generalized attack formulation that captures the "partial control" of the attacker. This formulation allows for stealthier attacks and extends the concept of action adversarial RL to more generalized settings. **(2)** We address the issue of non-convergence in adversarial training within our attack framework. By incorporating the principle of timescale separation, we achieve provable defenses and ensure theoretical guarantees for convergence to the globally most robust policy. **(3)** Empirical results affirm the efficacy of our generalized attack formulation in minimizing state distribution shifts and generating stealthier behaviors, compared to baseline unconstrained methods. Additionally, in tasks like Kuhn Poker and Robosumo, our timescale-separated adversarial training demonstrates superior stability and robustness when compared to popular baselines, including single-timescale adversarial training, self-play, and fictitious-play.

## 2 PRELIMINARIES

The extension of Markov decision processes (MDPs) with more than one agent is commonly modeled as Markov games (Littman, 1994). A Markov game with $N$ agents is defined by a tuple $\mathcal{G} = < N, \mathcal{S}, \{\mathcal{A}_i\}_{i=1}^N, P, \{r_i\}_{i=1}^N, \rho, \gamma >$, where $\mathcal{S}$ denotes the state space and $\mathcal{A}_i$ is the action space for agent $i$. The function $P$ controls the state transitions by the current state and one action from each agent: $P : \mathcal{S} \times \mathcal{A}_1 \times \cdots \times \mathcal{A}_N \to \Delta(\mathcal{S})$, where $\Delta(\mathcal{S})$ denotes the set of probability distributions over the state space $\mathcal{S}$. Given the current state $s_t$ and the joint action $(a_1, \ldots, a_N)$, the transition probability to $s_{t+1}$ is given by $P(s_{t+1}|s_t, a_1, \ldots, a_N)$. The initial state is sampled from the initial state distribution $\rho \in \Delta(S)$. Each agent $i$ also has an associated reward function $r_i : \mathcal{S} \times \mathcal{A}_1 \times \cdots \times \mathcal{A}_N \to [0, 1]$, whose goal is to maximize the $\gamma$-discounted expected return $\mathbb{E}[\sum_{t=0}^\infty \gamma^t r_i(s_t, a_i^t, a_{-i}^t)]$, where $-i$ is a compact representation of all complementary agents of $i$.

In Markov games, each agent is equipped with a policy $\pi_i : \mathcal{S} \to \Delta(\mathcal{A}_i)$ in policy class $\Pi_i$ and the joint policy is defined as $\boldsymbol{\pi}(\mathbf{a}|s) = \Pi_{i=1}^N \pi_i(a_i|s)$. Specifically, the value function for the victim agent $\nu$ given joint policy $(\pi_\nu, \pi_\alpha)$ is defined by $V_s(\pi_\nu, \pi_\alpha) = \mathbb{E}_{\pi_\nu, \pi_\alpha} \left[ \sum_{t=0}^\infty \gamma^t r_\nu(s_t, \mathbf{a}_t) \mid s_0 = s \right]$, where agent $\nu$ attempts to maximize the value function and attacker aims to minimize it. We abuse the notation to use $V_\rho(\pi_\nu, \pi_\alpha) := \mathbb{E}_{s \sim \rho}[V_s(\pi_\nu, \pi_\alpha)]$. We further define state visitation, which reflects how often the policy visits different states in the state space.

**Definition 2.1.** *(Stationary State Visitation) Let $d_\rho^{\boldsymbol{\pi}} \in \Delta(\mathcal{S})$ denote the normalized distribution of state visitation by following the joint policy $\boldsymbol{\pi}$ in the environment: $d_\rho^{\boldsymbol{\pi}}(s) = (1 - \gamma)\mathbb{E}_{s_0 \sim \rho} \sum_{t=0}^\infty \gamma^t P^{\boldsymbol{\pi}}(s_t = s|s_0).$*

## 3 A GENERALIZED ATTACK FORMULATION

**Problem description.** For simplicity, we consider a multi-agent system with two agents, $\nu$ and $\alpha$, following policies $\widehat{\pi}_\nu$ and $\widehat{\pi}_\alpha$ respectively. The interactions between these agents can be cooperative,

competitive, or mixed. As motivated earlier, we consider the attack scenario as described in Gleave et al. (2019); Wu et al. (2021b); Guo et al. (2021), where the threat comes from an attacker controlling agent $\alpha$. This attacker deviates from $\widehat{\pi}_\alpha$ to an adversarial policy $\widetilde{\pi}_\alpha$, aiming to minimize the performance of the victim agent $\nu$. Correspondingly, the victim's goal is to develop a more robust policy $\pi_\nu$ in anticipation of such adversarial policies. The interaction between the attacker and the victim can thus be modeled as a zero-sum game, regardless of the initial relationship between the two agents.[2] This framework can also be extended to settings with more than two agents, where the attacker controls multiple agents and adopts a joint adversarial policy.

**Attack formulation.** Although such attacks can effectively exploit the victim, in many practical scenarios, unlike the attacks in Gleave et al. (2019); Wu et al. (2021b); Guo et al. (2021), the attacker may face resistance and achieve only *partial* control of agent $\alpha$, e.g., in a hijack scenario. Therefore, we propose a more generalized attack framework. Here, the attacker aims to manipulate agent $\alpha$ using an adversarial policy $\widetilde{\pi}_\alpha$, but may not fully control the agent, which can still follow its original benign policy $\widehat{\pi}_\alpha$ with probability $1 - \epsilon_\pi$ at each time step, where $\epsilon_\pi \in [0, 1]$. Formally, under these conditions, the attacker solves the following attack objective:

$$\min_{\widetilde{\pi}_\alpha} V_\rho(\widehat{\pi}_\nu, \pi_\alpha) \tag{3.1}$$

$$\text{s.t.} \quad \pi_\alpha(\cdot \,|\, s) = (1 - \epsilon_\pi)\widehat{\pi}_\alpha(\cdot \,|\, s) + \epsilon_\pi \widetilde{\pi}_\alpha(\cdot \,|\, s), \quad \forall s \in \mathcal{S}. \tag{3.2}$$

Objective 3.1 is a standard attack objective (Gleave et al., 2019; Guo et al., 2021; Wu et al., 2021b), focused on minimizing the value of the victim $\nu$. The additional constraint 3.2 captures the probability $\epsilon_\pi$ at which the attacker can control agent $\alpha$. When $\epsilon_\pi = 1$, the setting degenerates to full control, aligning with Gleave et al. (2019); Guo et al. (2021); Wu et al. (2021b). Conversely, at $\epsilon_\pi = 0$, no attack occurs. This probability, denoted as the attack budget, effectively models the resistance encountered by the attacker. Its suitability as a budget will be further connected to action adversarial RL later (Tessler et al., 2019).

**(a) Effects of the attack budget.** Our proposed attack budget effectively characterizes the victim's performance degradation, serving as a viable measure of agent vulnerability. To substantiate this, we note a key observation related to a standard discrepancy measure (Kakade & Langford, 2002; Schulman et al., 2015). Given the constraint in Equation 3.2, for any $\widetilde{\pi}_\alpha$, the inequality $D_{\text{TV}}^{\max}(\pi_\alpha \| \widehat{\pi}_\alpha) \leq \epsilon_\pi$ holds. Here, $D_{\text{TV}}^{\max}(\pi_\alpha \| \widehat{\pi}_\alpha)$ is defined as $\max_s D_{\text{TV}}(\pi_\alpha(\cdot|s) \| \widehat{\pi}_\alpha(\cdot|s))$, and $D_{\text{TV}}(p \| q) := \frac{1}{2}\sum_i |p_i - q_i|$. This observation allows us to establish an upper bound on the victim's performance under an attack budget of $\epsilon_\pi$.

**Proposition 3.1** (*Bounded policy discrepancy induces bounded value discrepancy*)**.** *For two policy pairs* $(\widehat{\pi}_\nu, \widehat{\pi}_\alpha)$ *and* $(\widehat{\pi}_\nu, \pi_\alpha)$ *such that* $D_{\text{TV}}^{\max}(\pi_\alpha \| \widehat{\pi}_\alpha) \leq \epsilon_\pi$, *the difference between the victim value can be bounded as:* $|V_\rho(\widehat{\pi}_\nu, \widehat{\pi}_\alpha) - V_\rho(\widehat{\pi}_\nu, \pi_\alpha)| \leq \frac{2\epsilon_\pi}{(1-\gamma)^2}$.

This establishes a link between the attack budget and $|V_\rho(\widehat{\pi}_\nu, \widehat{\pi}_\alpha) - V_\rho(\widehat{\pi}_\nu, \pi_\alpha)|$. Specifically, the value function inherently satisfies a global Lipschitz condition. This implies that the attacker needs a sufficiently large attack budget $\epsilon_\pi$ to cause significant degradation in performance. This contrasts with supervised learning attacks at test time, where small perturbations can result in large performance shifts. Although this is a worst-case upper bound and may not be tight — especially when $\gamma$ is close to 1 — it still indicates that a longer effective game horizon grants the attacker greater capacity to degrade the victim's performance.

**(b) Attack model's stealthiness and detectability.** While the unconstrained attack in Gleave et al. (2019) significantly impairs the victim's performance, its overt nature makes it easy to detect even through static images. This deviates from the stealthy ethos of adversarial attacks in supervised learning (Goodfellow et al., 2014). In contrast, our generalized attack framework allows for partial control of an agent and enables stealthier attacks by regulating the attack budget $\epsilon_\pi$. Specifically, leveraging insights that static images alone can reveal attacks, we use generative modeling techniques for distribution matching to align the state distributions $d_\rho^{\widehat{\pi}_\nu, \widehat{\pi}_\alpha}$ and $d_\rho^{\widehat{\pi}_\nu, \pi_\alpha}$ induced by $(\widehat{\pi}_\nu, \widehat{\pi}_\alpha)$ and $(\widehat{\pi}_\nu, \pi_\alpha)$ respectively. Though exact state visitation is difficult to compute, regulating $\epsilon_\pi$ allows us manage the discrepancy between these distributions. We adopt total variation distance as our measure of discrepancy, offering the following guarantees.

---

[2]Even if $\widehat{\pi}_\nu$ and $\widehat{\pi}_\alpha$ are trained competitively, successful attacks can still occur due to sub-optimality of training Gleave et al. (2019); Bansal et al. (2017).

**Proposition 3.2** (*Bounded policy discrepancy induces bounded state distribution discrepancy*). *Fix any $\epsilon_\pi \in [0,1]$. For two policy pairs $(\widehat{\pi}_\nu, \widehat{\pi}_\alpha)$ and $(\widehat{\pi}_\nu, \pi_\alpha)$ such that $D_{\mathrm{TV}}^{\max}(\pi_\alpha || \widehat{\pi}_\alpha) \le \epsilon_\pi$, the discrepancy between the state distributions can be bounded as: $||d_\rho^{\widehat{\pi}_\nu, \widehat{\pi}_\alpha} - d_\rho^{\widehat{\pi}_\nu, \pi_\alpha}||_1 \le \frac{2\gamma\epsilon_\pi}{1-\gamma}$.*

Proposition 3.2 demonstrates that as long as $\epsilon_\pi$ is sufficiently small, the state distribution is well preserved, thus yielding images that are visually more similar to the original ones. This suggests that in practice, $\epsilon_\pi$ can be treated as a hyper-parameter, balancing the attacker's performance with stealthiness. Finally, comparing actions or rewards is also a viable method to detect potential attacks. However, in many practical multi-agent systems, agents are decentralized, and actions or rewards are private to each agent (Zhang et al., 2018b), not always available to humans aiming to detect potential adversarial attacks.

*Comparison with single-agent stealthy attacks.* Russo & Proutiere (2021); Franzmeyer et al. (2022) consider stealthy attacks in a *different* setting, involving adversarial state or action perturbations, within single-agent RL. Their concept of unstealthiness or detectability is predicated on the inconsistencies in the *transition dynamics* when states or actions are adversarially perturbed, necessitating an accurate world model. However, in our scenario, even if such a world model is accessible, the (global) transition dynamics $P$ remain unaffected by the adversarial policy. Concurrently, comparing the *marginalized* transition dynamics induced by $\widehat{\pi}_\alpha$ and $\pi_\alpha$ is plausible *from the perspective of the victim $\nu$*. Based on Proposition 3.3, inconsistencies in the marginalized transition dynamics can also be upper-bounded by the variation in the policy space, assuring low *detectability* as considered in Russo & Proutiere (2021); Franzmeyer et al. (2022). Thus, even if humans or detectors can access the private actions of the agent $\nu$ and establish accurate corresponding marginalized transition dynamics, discrepancies might remain undetected as long as $\epsilon_\pi$ is maintained minimal.

**Proposition 3.3** (*Bounded policy discrepancy induces bounded marginalized transition dynamics inconsistencies*). *We define the marginalized transition dynamic of agent $\nu$ as $P_\nu^{\pi_\alpha}(s' \,|\, s, a_\nu) := \mathbb{E}_{a_\alpha \sim \pi_\alpha(\cdot \,|\, s)}[P(s' \,|\, s, a_\alpha, a_\nu)]$ for given $\pi_\alpha$. $P_\nu^{\widehat{\pi}_\alpha}$ is defined similarly for the policy $\widehat{\pi}_\alpha$. Then for any $s \in \mathcal{S}$ and $a_\nu \in \mathcal{A}_\nu$, we have $D_f\left(P_\nu^{\pi_\alpha}(\cdot \,|\, s, a_\nu) \,||\, P_\nu^{\widehat{\pi}_\alpha}(\cdot \,|\, s, a_\nu)\right) \le D_f\left(\pi_\alpha(\cdot \,|\, s) \,||\, \widehat{\pi}_\alpha(\cdot \,|\, s)\right)$, where $D_f$ is any $f$-divergence, which includes $D_{\mathrm{TV}}$, connecting back to the attack budget.*

**(c) Connection to action adversarial RL.** Intriguingly, our attack formulation also extends the single-agent action adversarial RL (Tessler et al., 2019) to a multi-agent setting. Specifically, in PR-MDP (cf. Definition 1 of Tessler et al. (2019)), the policy under attack aligns with our Equation 3.2. In the context of single-agent action adversarial RL, the policy $\widetilde{\pi}_\alpha$ is only a *part* of the finally executed policy, while the policy $\widehat{\pi}_\alpha$ represents the victim. Thus, our formulation broadens the attack setting of Tessler et al. (2019) to multi-agent RL, considering the *other* agent $\nu$ as the victim instead of the agent $\alpha$ itself. Moreover, determining the most robust policy for the victim using the policy iteration scheme for PR-MDP from Tessler et al. (2019) becomes inefficient in our context due to the absence of specific structures inherent in PR-MDP (Section 4 of Tessler et al. (2019)).

Henceforth, we will abbreviate $\widetilde{\pi}_\alpha$ as $\pi_\alpha$ without ambiguity, and the actually deployed policy for agent $\alpha$ is represented as $(1 - \epsilon_\pi)\widehat{\pi}_\alpha + \epsilon_\pi\pi_\alpha$. Detailed proofs and discussions related to this section are available in §D.

## 4 Improved adversarial training with timescale separation

**On the necessity and challenge of provably efficient defenses.** As discussed before, to provide effective defenses, there are unique challenges standing out compared with single-agent robust RL (Tessler et al., 2019). Meanwhile, finding the celebrated solution concept Nash Equilibrium (NE) between the attacker and the victim suffices for finding the most robust victim policy during robust training but may not be necessary since NE guarantees that the attacker is also non-exploitable. We provide more detailed discussions on the relationship between NE and robustness in §B. There are a bunch of existing works solving NE for structured extensive-form games (Lockhart et al., 2019; Brown et al., 2019; Sokota et al., 2022) or for general games but without provably efficient guarantees (Fudenberg & Levine, 1995; Lanctot et al., 2017; Balduzzi et al., 2019; Muller et al., 2019). In practice, general game-theoretical methods often require solving best response problems iteratively, thus being computationally expensive. In theory, simply plugging in *black-box* NE solvers

may not solve our problem with provable efficiencies since finding even *local* NE for a *general* nonconvex-nonconcave problem is computationally hard (Daskalakis et al., 2021). Therefore, instead of adopting a black-box game-theoretical solver, we investigate adversarial training, a popular and more efficient paradigm for robust RL (Pinto et al., 2017; Zhang et al., 2021a; Sun et al., 2021).

There are prior works that utilize well-trained attacks for re-training to fortify the robustness of the victim (Gleave et al., 2019; Guo et al., 2021; Wu et al., 2021b). However, it has been demonstrated that while re-training against a specific adversarial policy does augment robustness against it, the performance against other policies may be compromised as validated by Gleave et al. (2019). Intuitively, if the victim is retrained against a specific attacker, its policy might be overfitted to that attacker. Thus, it is vital to uphold the performance of the victim against all potential attackers. Rather than merely re-training against a specific attacker, adversarial training methods have been shown to be effective in bolstering robustness against a broad spectrum of adversarial attacks. In these methods, the victim and the attacker are trained alternatively or simultaneously (Zhang et al., 2020b; Pinto et al., 2017). Here, we re-examine adversarial training in the RL domain and demonstrate that prevalent adversarial training methods encounter a *non-converging* problem with either alternative or simultaneous training, for which we defer examples and detailed discussions to §C. To address these issues formally, we contemplate the robustness of the victim and define the exploitability of $\pi_\nu$ under the *worst-case* attack as follows.

**Definition 4.1** ((One-side) exploitability). *Given $\epsilon_\pi \in [0,1]$ and $\widehat{\pi}_\alpha$, for a victim policy $\pi_\nu$, we measure the robustness of $\pi_\nu$ by:* $\mathrm{Expl}(\pi_\nu) = -\min_{\pi_\alpha} V_\rho(\pi_\nu, (1-\epsilon_\pi)\widehat{\pi}_\alpha + \epsilon_\pi \pi_\alpha)$.

**Intuitions of timescale separation.** The smaller $\mathrm{Expl}(\pi_\nu)$ is, the more robust $\pi_\nu$ is. Therefore, to ensure the worst-case performance against the strongest adversarial policy, the victim should optimize the policy according to $\min_{\pi_\nu} \mathrm{Expl}(\pi_\nu)$. Ideally, if we can derive an analytical form of the function $\mathrm{Expl}(\cdot)$ or compute its gradient, then we can simply run gradient descent to optimize it. Unfortunately, it is not obvious how to derive an analytical form and the function may not be even differentiable, let alone computing the gradient since the function relies on solving a minimization problem. However, it is possible to first solve the minimization problem, getting $\pi_\alpha^\star$, and compute the gradient w.r.t $\pi_\nu$, namely $\nabla_{\pi_\nu} - V_\rho(\pi_\nu, (1-\epsilon_\pi)\widehat{\pi}_\alpha + \epsilon_\pi \pi_\alpha^\star)$, *as if $\pi_\alpha^\star$ is fixed, hoping it could serve as a good descent direction*. Formalizing this intuition, we propose to improve adversarial training via *timescale separation* with Min oracle (shown in Algorithm 1), where timescale separation comes from the fact the attacker takes a min step against the victim in line 3 while the victim takes only one gradient update in line 4. Note Lockhart et al. (2019) also considers *directly* minimizing the exploitability function but the algorithm and analysis are only applicable to extensive-form games. Finally, we remark that Algorithm 1 is consistent with the leader-follower update style that is developed for Stackelberg equilibrium in multi-agent RL (Gerstgrasser & Parkes, 2023).

---

**Algorithm 1** Adversarial Training with Min-oracle

1: **Input:** random policy $\pi_\nu^0$, learning rate sequence $\{\eta^t\}$
2: **for** $t = 0$ **to** $T$ **do**
3:     $\pi_\alpha^t \leftarrow \arg\min_{\pi_\alpha} V_\rho(\pi_\nu^t, (1-\epsilon_\pi)\widehat{\pi}_\alpha + \epsilon_\pi \pi_\alpha)$.
4:     $\pi_\nu^{t+1} \leftarrow \mathcal{P}_{\Pi_\nu}(\pi_\nu^t + \eta^t \nabla_{\pi_\nu} V_\rho(\pi_\nu^t, (1-\epsilon_\pi)\widehat{\pi}_\alpha + \epsilon_\pi \pi_\alpha^t))$. // projection onto the simplex
5: **Output:** sample $\pi_\nu^t$ with probability proportional to $\eta^t$.

---

**Efficient approximation.** The min oracle used in Algorithm 1 can be implemented with standard RL algorithms like PPO. When the game has special structures like extensive-form games (Lockhart et al., 2019) or one agent has a substantially smaller state/action space, such min oracle can be even implemented quite efficiently. However, in general, to make one gradient update for agent $\nu$, agent $\alpha$ needs to compute a complete best response, which is computationally expensive in practice. To fix this issue, we utilize the idea of using a much faster update scale for agent $\alpha$ so that when agent $\nu$ performs the gradient update, the policy $\pi_\alpha^t$ is always and already an *approximate solution* of $\arg\min_{\pi_\alpha} V_\rho(\pi_\nu^t, (1-\epsilon_\pi)\widehat{\pi}_\alpha + \epsilon_\pi \pi_\alpha)$. Formally, in addition to Algorithm 1, we present an alternative efficient Algorithm 2, where the min oracle is replaced by a gradient update with a larger step size and both agents only need to perform a gradient update independently. *Therefore, our final algorithm is simple and compatible with standard RL algorithms, like PPO to implement the gradient update step for both agents, avoiding solving best responses at each iteration under popular*

*game-theoretical approaches (Fudenberg & Levine, 1995; Lanctot et al., 2017; Balduzzi et al., 2019; Muller et al., 2019).* To validate our intuitions and verify that our algorithms do provide provably efficient defenses, we shall prove the convergence guarantee of both algorithms in the next section.

---

**Algorithm 2** Adversarial Training with Two Timescales

---

1: **Input:** random policy $\pi_\nu^0$, $\pi_\alpha^0$, learning rate sequence $\{\eta_\nu^t\}$, $\{\eta_\alpha^t\}$, such that $\eta_\nu^t \ll \eta_\alpha^t$.
2: **for** $t = 0$ **to** $T$ **do**
3:     $\pi_\alpha^{t+1} \leftarrow \mathcal{P}_{\Pi_\alpha}(\pi_\alpha^t - \eta_\alpha^t \nabla_{\pi_\alpha} V_\rho(\pi_\nu^t, (1 - \epsilon_\pi)\widehat{\pi}_\alpha + \epsilon_\pi \pi_\alpha^t))$.
4:     $\pi_\nu^{t+1} \leftarrow \mathcal{P}_{\Pi_\nu}(\pi_\nu^t + \eta_\nu^t \nabla_{\pi_\nu} V_\rho(\pi_\nu^t, (1 - \epsilon_\pi)\widehat{\pi}_\alpha + \epsilon_\pi \pi_\alpha^t))$.
5: **Output:** sample $\pi_\nu^t$ with probability proportional to $\eta_\nu^t$.

---

## 5 THEORETICAL ANALYSIS

To understand and verify our approaches, we start by considering direct policy parameterization for both agents $\nu$ and $\alpha$, which is already challenging due to the non-convexity, non-concavity, and serves as the first step to analyze more complex function approximations.

**Definition 5.1** (Direct parameterization). *The policies $\pi_\nu$ and $\pi_\alpha$ have the parameterization $\pi_\nu(a \,|\, s) = \nu_{s,a}$, $\pi_\alpha(a \,|\, s) = \alpha_{s,a}$, where $\nu \in \Delta(\mathcal{A}_\nu)^{|\mathcal{S}|}$ and $\alpha \in \Delta(\mathcal{A}_\alpha)^{|\mathcal{S}|}$.*

For convenience of discussions, given $\epsilon_\pi$ and $\widehat{\pi}_\alpha$, we will write $J_{\epsilon_\pi}(\pi_\nu, \pi_\alpha) := V_\rho(\pi_\nu, (1 - \epsilon_\pi)\widehat{\pi}_\alpha + \epsilon_\pi \pi_\alpha)$. Before proving the convergence of our methods, we define the mismatch coefficient to measure the intrinsic hardness of the environment. This is achieved by comparing the stationary state occupancy frequencies under certain policies against the initial state distribution. In simpler terms, a smaller value of this quantity indicates that the environment is more easily explorable.

**Definition 5.2.** *Given the Markov game $\mathcal{G}$, benign policy $\widehat{\pi}_\alpha$, and attack budget $\epsilon_\pi$, we define the minimax mismatch coefficient as*

$$C_\mathcal{G}^{\epsilon_\pi} := \max \left\{ \max_{\pi_\nu \in \Pi_\nu} \min_{\pi_\alpha \in \Pi_\alpha^\star(\pi_\nu)} \left\| \frac{d_\rho^{\pi_\nu, (1 - \epsilon_\pi)\widehat{\pi}_\alpha + \epsilon_\pi \pi_\alpha}}{\rho} \right\|_\infty , \max_{\pi_\alpha \in \Pi_\alpha} \min_{\pi_\nu \in \Pi_\nu^\star(\pi_\alpha)} \left\| \frac{d_\rho^{\pi_\nu, (1 - \epsilon_\pi)\widehat{\pi}_\alpha + \epsilon_\pi \pi_\alpha}}{\rho} \right\|_\infty \right\},$$

*where $\Pi_\alpha^\star(\pi_\nu) := \arg\min_{\pi_\alpha \in \Pi_\alpha} J_{\epsilon_\pi}(\pi_\nu, \pi_\alpha)$, and $\Pi_\nu^\star(\pi_\alpha) := \arg\max_{\pi_\nu \in \Pi_\nu} J_{\epsilon_\pi}(\pi_\nu, \pi_\alpha)$.*

With those two definitions, we can analyze how the robustness of the victim improves during adversarial training as follows:

**Theorem 5.3.** *Fix any $\delta > 0, \epsilon_\pi \in [0, 1]$. For Algorithm 1, suppose the learning rate $\eta_\nu^t \asymp \delta$, after $T$ iterations, it is guaranteed that $\frac{1}{T} \sum_{t=1}^{T} \mathrm{Expl}(\pi_\nu^t) \leq \min_{\pi_\nu} \mathrm{Expl}(\pi_\nu) + \delta$, where $T = \frac{1}{\delta^2} \mathrm{poly}(C_\mathcal{G}^{\epsilon_\pi}, |\mathcal{S}|, |\mathcal{A}_\alpha|, |\mathcal{A}_\nu|, \frac{1}{1-\gamma})$; while for Algorithm 2, suppose the learning rate $\eta_\nu^t \asymp \delta^8$, $\eta_\alpha^t \asymp \delta^4$, after $T$ iterations, it is guaranteed that $\frac{1}{T} \sum_{t=1}^{T} \mathrm{Expl}(\pi_\nu^t) \leq \min_{\pi_\nu} \mathrm{Expl}(\pi_\nu) + \delta$, where $T = \mathrm{poly}(\frac{1}{\delta}, C_\mathcal{G}^{\epsilon_\pi}, |\mathcal{S}|, |\mathcal{A}_\alpha|, |\mathcal{A}_\nu|, \frac{1}{1-\gamma})$.*

**Remark 5.4.** *To get a non-vacuous finite time convergence, the mismatch coefficient needs to be bounded, which is standard and necessary in the analysis of policy gradient methods (Daskalakis et al., 2020; Agarwal et al., 2021), where our definition of $C_\mathcal{G}^{\epsilon_\pi}$ is based on the definition in (Daskalakis et al., 2020). It is worth noticing that such an assumption is weaker than other similar notions such as concentrability (Munos, 2003; Chen & Jiang, 2019), without requiring every visitable state to be visited at the first time step.*

**Implications.** This theorem demonstrates that, *on average*, the victim policy $\pi_\nu^t$ is assured to converge to the *most robust* one; that is, the solution of $\arg\max_{\pi_\nu} \min_{\pi_\alpha} V_\rho(\pi_\nu, (1 - \epsilon_\pi)\widehat{\pi}\alpha + \epsilon\pi\pi_\alpha)$. Theorem 5.3 reveals that Algorithm 1 achieves better iteration complexity owing to a larger learning rate. Meanwhile, the convergence for Algorithm 2 also substantiates the necessity of timescale separation due to $\eta_\nu^t \ll \eta_\alpha^t$. To the best of our knowledge, the analysis for Algorithm 1 is new, even for the $\epsilon_\pi = 1$ case, and the analysis for Algorithm 2 leverages Daskalakis et al. (2020); Jin et al. (2020);

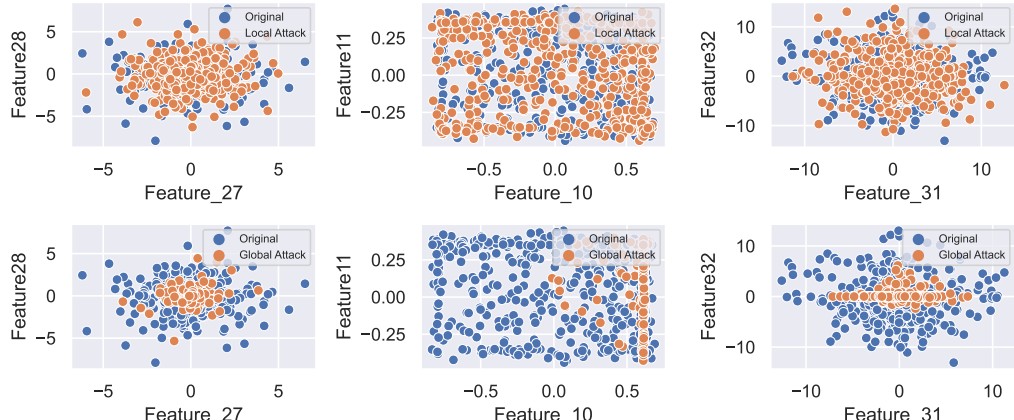

Figure 1: Visualization and comparison of our proposed constrained attack with $\epsilon_\pi = 0.2$ (first row) vs. an unconstrained attack (second row, $\epsilon_\pi = 1$), under the condition that both achieve the same attacking success rate. The most important state features are shown. It is clear that our constrained adversarial policy induces much smaller state distribution shifts.

Zhang et al. (2021b) but addresses the more prevalent discounted reward setting *without* assuming the game to cease at every state with a positive probability. It also permits the attacker to exert only partial control over the agent $\alpha$ within any stipulated attack budget $\epsilon_\pi$. Importantly, our theorem can be readily extended to encompass results of *last-iterate* convergence using the regularization techniques highlighted in Zeng et al. (2022) and *stochastic* gradients by deploying popular gradient estimators from finite samples, with the central concept remaining timescale separation. Detailed proofs are available in §D.

## 6 RELATED WORK

**Stealthy adversarial attacks in RL.** To render attacks on RL policies more feasible and practical, Sun et al. (2020) demonstrates that targeting critical points can facilitate efficient and stealthy attacks. Russo & Proutiere (2021) optimizes both attack detectability and victim performance, analyzing the trade-off between them. Franzmeyer et al. (2022) introduces an illusionary attack that maintains consistency with the environment's transition dynamics, necessitating a world model. The aforementioned stealthy attacks focus on perturbing the victim's state observations/actions. In contrast, this paper presents a generalized attack framework for multi-agent systems, allowing for stealth by managing the attack budget, wherein the attacker indirectly influences the victim by altering another agent's policy.

**Timescale separation for adversarial training.** Adversarial training is a widely-adopted method for cultivating models robust against adversarial attacks. The efficacy of timescale separation in this context has been empirically affirmed; having increased loops for the inner attack subroutine translates to enhanced robustness Madry et al. (2017); Shafahi et al. (2019). Likewise, in training GANs (Heusel et al., 2017), utilizing a larger learning rate for the discriminator surpasses conventional GAN training and ensures convergence to a local NE. Moreover, Fiez & Ratliff (2021) explores more general non-convex non-concave zero-sum games and elucidates the local convergence to strict local minimax equilibrium with finite timescale separation. Contrarily, our adversarial training algorithm is assured to converge to the (globally) most robust policy.

## 7 EXPERIMENTS

Our experiments utilize two standard environments: Kuhn Poker (Kuhn, 1950; Lanctot et al., 2019) and RoboSumo (Al-Shedivat et al., 2017). Detailed introductions to these environments, implementation specifics, and hyper-parameters are outlined in §E. Unless specified otherwise, the results are averaged over 5 seeds. In this section, we seek to address the following pivotal questions:

▷ **Q1.** Can our generalized attack formulation produce fewer state distribution variations and more stealthy behaviors compared to an unconstrained attack?

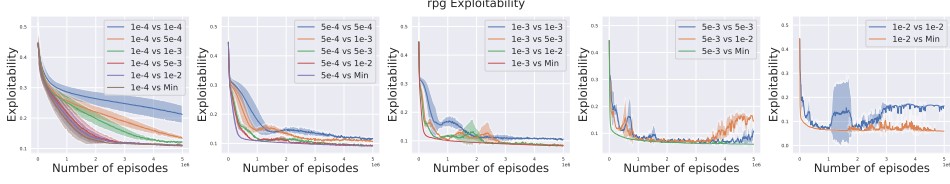

(a) Exploitability of the victim when using RPG for policy gradient.

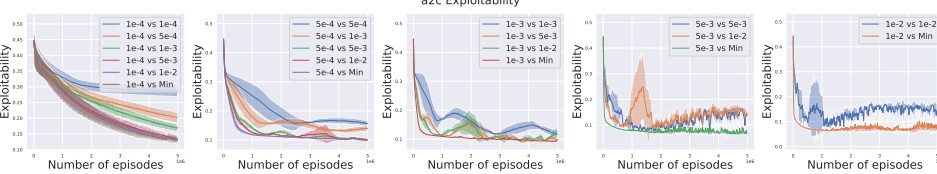

(b) Exploitability of the victim when using A2C for policy gradient.

Figure 2: Exploitability of victim policy in Kuhn Poker trained by two timescale and single timescale (min indicates the policy trained with a min oracle).

▷ **Q2.** Will adversarial training with timescale separation (involving a min oracle and two timescales) exhibit more stable learning dynamics, and can the two timescales algorithm effectively approximate the adversarial training with a min oracle?

▷ **Q3.** In complex environments, where a min oracle may not be available, can adversarial training with two timescales enhance robustness compared to prevalent and acclaimed baselines?

**Controllable adversarial attack (Q1).** We conduct experiments on the Robosumo environment. To verify that our attack formulation indeed achieves smaller state distribution variations, we choose an unconstrained adversarial policy and a constrained policy with the same winning rate for fair comparisons. Specifically, we investigate the distribution shift in the victim's observation part of the state features.

To select essential state features, we employ the variance-based feature importance method to filter out state features with small variances as they are deemed unimportant. The sorted feature importance for the RoboSumo games is depicted in Figure E.2. Figure 1 demonstrates that the adversarial policy, derived from our generalized attack framework, induces a much smaller state distribution shift compared to the unconstrained adversarial policy when assessed under the same winning rate. Additionally, we quantify the state-distribution shift brought about by the constrained and unconstrained attacks by calculating the Wasserstein-2 distance between their state distributions, as illustrated in Figure 3. **Our constrained attack results in a significantly lower state distribution shift compared to the unconstrained one.** To confirm that our generalized attack methods, with a regulated attack budget, do indeed produce more stealthy behaviors,

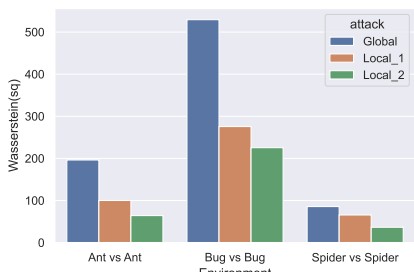

Figure 3: State-distribution shift w.r.t Wasserstein-2 distance (squared) incurred due to the Global ($\epsilon_\pi = 1$), Local_1 ($\epsilon_\pi = 0.7$), Local_2($\epsilon_\pi = 0.3$) attacks in 3 Robosumo environments.

we visualize agents with $\epsilon_\pi \in 0.3, 0.7, 1$; **here, a smaller $\epsilon_\pi$ induces behaviors that are visually more similar to the system without an attack** (see gifs at https://sites.google.com/view/stealthy-attack). An ablation study on the trade-off between stealthiness and the attacker's performance is also presented in §E.4, also validating the effectiveness of $\epsilon_\pi$ in regulating the attacker's strength, in line with Proposition 3.1. We show in §E.4, **even when there is a mismatch for attack budgets between training time and test time, the victim policy still shows greatly improved robustness.**

**Adversarial Training with Timescale Separation (Q2&3).** To address Q2&3, we examine the learning dynamics and robustness of policies trained with and without timescale separation, comparing these to other baselines in both Kuhn Poker and Robosumo environments.

*Kuhn Poker.* We implement Algorithm 1 and 2 under OpenSpiel (Lanctot et al., 2019), with the min oracle achieved through game tree search. For gradient update, we utilize Regret Policy Gradient (RPG) (Srinivasan et al., 2018) and Advantage Actor-Critic (A2C) (Mnih et al., 2016) in lieu of the vanilla policy gradient. Figure 2 illustrates the exploitability of the victim policy $\pi_\nu$, **where adver-**

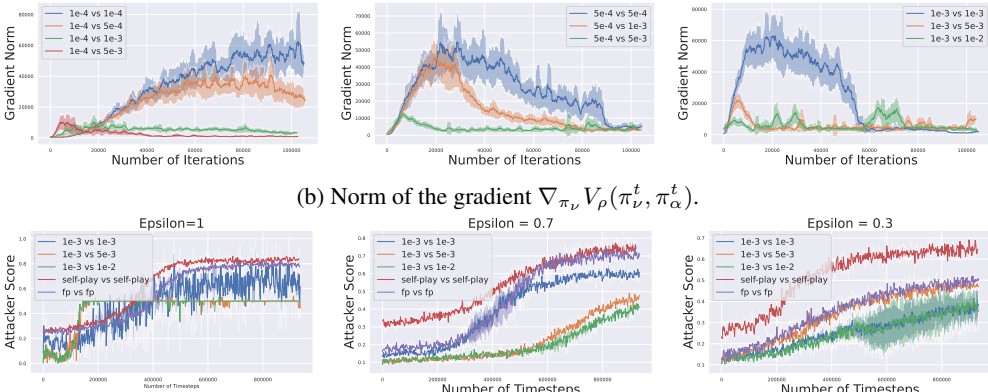

(a) Score of the victim policy, which is computed by winning rate + tie rate/2. Note here different from other plots, to show the potential oscillation behaviors during training, we show just one seed instead of multiple ones.

(b) Norm of the gradient $\nabla_{\pi_\nu} V_\rho(\pi_\nu^t, \pi_\alpha^t)$.

(c) Score of the attacker against robustified policy (final output of adversarial training). Lower is better.

Figure 4: (a). The score of the victim policy trained with two timescales converges rapidly, while the policy trained only with a single timescale suffers from much more oscillations. (b). The gradient norm trained by two timescales is also much smaller. (c). Under different $\epsilon_\pi = 1, 0.7, 0.3$, when attacking the robustified victim policy, i.e. computing $\min_{\pi_\alpha} V_\rho(\pi_\nu^\star, (1-\epsilon_\pi)\widehat{\pi}+\epsilon_\pi\pi_\alpha)$ with standard RL algorithm, victim trained by two timescales achieves the lowest exploitability/best robustness.

**sarial training with the min oracle exhibits the fastest convergence, lowest exploitability, and least variance.** Meanwhile, policies trained with a sufficient timescale separation parameter $\eta_\alpha^t/\eta_\nu^t$ closely approximate the algorithm with a min oracle, outperforming single timescale algorithms.

*Robosumo Competition.* To demonstrate the scalability of timescale separation, we evaluate our methods on Robosumo—a high-dimensional, continuous control task, representing a significant challenge in terms of both training and evaluation. Although a min oracle with game tree search is unattainable in such a continuous control task, earlier experiments suggest that an ample timescale separation ratio can effectively approximate it. We monitor the score, $V_\rho(\pi_\nu^t, (1-\epsilon_\pi)\widehat{\pi}_\alpha+\epsilon_\pi\pi_\alpha^t)$, and the norm of gradient $\nabla_{\pi_\nu} V_\rho(\pi_\nu^t, (1-\epsilon_\pi)\widehat{\pi}_\alpha + \epsilon_\pi\pi_\alpha^t)$ during adversarial training. As per Figure 4a and 4b, we ascertain that **single timescale training results in unstable behaviors, large variance, and gradient norm, while the two timescale training achieves quick convergence and significantly smaller gradient norm.** Crucially, to affirm the robustness of our methods, we compare them with single timescale adversarial training, self-play (Bansal et al., 2017), and fictitious-play (Heinrich & Silver, 2016). Self-play and fictitious-play-based methods are renowned for training adversarially robust RL agents (Pinto et al., 2017; Zhang et al., 2021a; Tessler et al., 2019). We employ PPO to calculate the best response of the finalized robustified victim policy, illustrating the performance of the attacker during the victim attack process in Figure 4c. Here, a lower winning rate for the attacker signifies enhanced robustness. **This demonstrates that our adversarial training with two timescales leads to victim policies with enhanced robustness compared to popular baseline methods, single timescale adversarial training, self-play, and fictitious-play.**

## 8 DISCUSSION AND LIMITATIONS

In this paper, we reassess the threats posed to RL agents by adversarial policies by introducing a generalized attack formulation and develop the first provably efficient defense algorithm, "adversarial training with timescale separation", with convergence to the most robust policy under mild conditions. Meanwhile, we leave how to scale our formulation to accommodate multiple independent attackers with self-interested adversarial policies as future works.

## 9 ACKNOWLEDGEMENT

Liu, Chakraborty, Sun, and Huang are supported by National Science Foundation NSF-IIS-FAI program, DOD-ONR-Office of Naval Research, DOD Air Force Office of Scientific Research, DOD-DARPA-Defense Advanced Research Projects Agency Guaranteeing AI Robustness against Deception (GARD), Adobe, Capital One and JP Morgan faculty fellowships. The authors also thank Kaiqing Zhang and Soheil Feizi for the valuable discussions at the initial stages.

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

# Appendix for "Rethinking Adversarial Policies: A Generalized Attack Formulation and Provable Defense in RL"

## Table of Contents

## A   ADDITIONAL RELATED WORK

**Two-player zero-sum games.** The interaction between the attacker and the victim can be modeled as a two-player zero-sum game. There is a large body of work using RL to solve Nash equilibrium (NE). For example, Sokota et al. (2022); Perolat et al. (2021); McAleer et al. (2021); Brown et al. (2019); Lockhart et al. (2019) considers the normal-form and extensive-form games. Heinrich & Silver (2016); Lanctot et al. (2017); McAleer et al. (2020); Perez-Nieves et al. (2021); Liu et al. (2021) deal with more general two-player zero-sum games and propose population-based RL algorithms showing empirical success but lacking provable finite-time guarantees. Daskalakis et al. (2020); Zeng et al. (2022) analyze the theory of independent gradient with different learning rates in *Markov game*, which is a *special case* of our defense problem against *unconstrained attack* and serves as the inspiration for us to develop provable adversarial training algorithms.

**Adversarial attacks on RL policies.** As deep neural networks are shown to be vulnerable to adversarial attacks (Szegedy et al., 2014; Goodfellow et al., 2015), the adversarial robustness of deep RL policies has also attracted increasing attention. One of the earliest works by Huang et al. (2017) reveals the vulnerability of neural policies by adapting various adversarial attacks from supervised learning to RL policies. Lin et al. (2019) consider an efficient attack that only perturbs the agent at a subset of time steps. There has been subsequent research in developing stronger pixel-based attacks (Qiaoben et al., 2021; Pattanaik et al., 2017; Oikarinen et al., 2020). Zhang et al. (2020b) built the theoretical framework SA-MDP for adversarial state perturbation and proposed the corresponding regularizer for more robust reinforcement learning policies. Subsequent work by Sun et al. (2021) improves Zhang et al. (2020b) with the framework of PA-MDP for better efficiency. The majority of related work on adversarial RL focuses on perturbing state observations (Huang et al., 2017; Oikarinen et al., 2020; Sun et al., 2021), and assumes that the perturbation is small in $\ell_p$ distance. In contrast, our paper considers the attack generated by other agents in a multi-agent system and does not restrict the perturbation distance in every single step, allowing for more flexible and practical attack models.

**Adversarial attacks on multi-agent RL (MARL).** Gleave et al. (2019) investigate adversarial policies in a two-player zero-sum game, where a victim can be exploited and significantly misled by the opponent's changed behavior. Guo et al. (2021) remove the zero-sum assumption in Gleave et al.

(2019) and construct a new formulation for the adversarial policy. Lin et al. (2020) study adversarial attacks in cooperative MARL systems and reveal that attacking one of the agents in the team can greatly reduce the total reward. However, these adversarial policies are unconstrained and could cause abnormal behaviors that are easily detectable, while our attack model can be made stealthy by restricting the state distribution shifts. We also propose provably robust defense algorithms to learn a policy that is unexploitable.

**Attacks and defenses on communication in MARL.** There is also a line of work studying attacks and defenses on communications in MARL (Blumenkamp & Prorok, 2020; Tu et al., 2021; Mitchell et al., 2020; Xue et al., 2022), where the communication among cooperative agents could be perturbed to influence the decisions of victims. However, we consider adversarial behaviors or policies of other agents, which affect the victim by the actions taken by other agents.

**Provably robust defenses.** To provide guaranteed robustness for deep neural networks, many approaches have been developed to certify the performance of neural networks, including semidefinite programming-based defenses (Raghunathan et al., 2018a;b), convex relaxation of neural networks (Gowal et al., 2019; Zhang et al., 2018a; Wong & Kolter, 2018; Zhang et al., 2020a; Gowal et al., 2018), randomized smoothing of a classifier (Cohen et al., 2019; Hayes, 2020), etc. In an effort to certify RL agents' robustness, some approaches (Lütjens et al., 2020; Zhang et al., 2020b; Oikarinen et al., 2020; Fischer et al., 2019) apply network certification tools to bound the Q networks. Kumar et al. (2021) and Wu et al. (2021a) apply randomized smoothing (Cohen et al., 2019) to RL to achieve provable robustness. These defenses mainly focus on the adversarial perturbations directly applied to the agent's inputs. Sun et al. (2022) propose a certifiable defense against adversarial communication in MARL systems. To the best of our knowledge, our paper is the first to provide provable convergence guarantees for adversarial training against adversarial attacks on the behaviors of other agents in the environment.

**Improving policy robustness by adversarial training.** Prior work shows that the competition between the victim agent and the adversary can be regarded as a two-player zero-sum game while training agents with learned adversarial attacks can improve the robustness of the victim. Such an adversarial training paradigm has been shown effective under state perturbations (Zhang et al., 2021a; Sun et al., 2021; Liang et al., 2022) and action perturbations (Pinto et al., 2017; Tessler et al., 2019) on a single victim. In the context of perturbing the actions of the victim, Tessler et al. (2019) presents a two-player policy iteration algorithm that is proved to converge to the Nash Equilibrium. In contrast, we consider the adversarial behaviors of other agents, and we provide both theoretical guarantees and empirical evidence for the effectiveness of our adversarial training.

## B  RELATIONSHIP BETWEEN NE AND ROBUSTNESS

To understand the relationship between NE and robustness, we formally define the NE as follows.

**Definition B.1** (Nash equilibrium). *Fix $\widehat{\pi}_\alpha$ and $\epsilon_\pi \geq 0$. We say a pair of policy $(\pi_\nu^\star, \pi_\alpha^\star)$ the Nash equilibrium for the zero-sum game between the victim and the attacker if it holds that for any $\pi_\nu'$ and $\pi_\alpha'$:*

$$V_\rho(\pi_\nu^\star, (1 - \epsilon_\pi)\widehat{\pi}_\alpha + \epsilon_\pi \pi_\alpha^\star) \geq V_\rho(\pi_\nu', (1 - \epsilon_\pi)\widehat{\pi}_\alpha + \epsilon_\pi \pi_\alpha^\star), \tag{B.1}$$

$$V_\rho(\pi_\nu^\star, (1 - \epsilon_\pi)\widehat{\pi}_\alpha + \epsilon_\pi \pi_\alpha^\star) \leq V_\rho(\pi_\nu^\star, (1 - \epsilon_\pi)\widehat{\pi}_\alpha + \epsilon_\pi \pi_\alpha'). \tag{B.2}$$

Now we have the following proposition showing the robustness of the NE.

**Proposition B.2.** *Fix $\widehat{\pi}_\alpha$ and $\epsilon_\pi \in [0, 1]$. If $(\pi_\nu^\star, \pi_\alpha^\star)$ is the Nash equilibrium for the zero-sum game between the victim and the attacker, then $\pi_\nu^\star$ is the minimizer for the function $\mathrm{Expl}(\pi_\nu')$.*

*Proof.* According to Definition B.1, it holds that

$$\min_{\pi_\alpha'} V_\rho(\pi_\nu^\star, (1 - \epsilon_\pi)\widehat{\pi}_\alpha + \epsilon_\pi \pi_\alpha') = V_\rho(\pi_\nu^\star, (1 - \epsilon_\pi)\widehat{\pi}_\alpha + \epsilon_\pi \pi_\alpha^\star) = \max_{\pi_\nu'} V_\rho(\pi_\nu', (1 - \epsilon_\pi)\widehat{\pi}_\alpha + \epsilon_\pi \pi_\alpha^\star).$$

$$\tag{B.3}$$

Combining Equation equation B.3 with the fact that for any function $f$ it holds that $\min_x \max_y f(x, y) \geq \max_y \min_x f(x, y)$, we conclude that

$$V_\rho(\pi_\nu^\star, (1 - \epsilon_\pi)\widehat{\pi}_\alpha + \epsilon_\pi \pi_\alpha^\star) = \max_{\pi_\nu'} \min_{\pi_\alpha'} V_\rho(\pi_\nu', (1 - \epsilon_\pi)\widehat{\pi}_\alpha + \epsilon_\pi \pi_\alpha')$$

$$= \min_{\pi_\alpha'} \max_{\pi_\nu'} V_\rho(\pi_\nu', (1 - \epsilon_\pi)\widehat{\pi}_\alpha + \epsilon_\pi \pi_\alpha').$$

Now given an NE pair $(\pi_\nu^\star, \pi_\alpha^\star)$, we consider the exploitability of $\pi_\nu^\star$:

$$\mathrm{Expl}(\pi_\nu^\star) = -\min_{\pi_\alpha} V_\rho(\pi_\nu^\star, (1 - \epsilon_\pi)\widehat{\pi}_\alpha + \epsilon_\pi \pi_\alpha)$$

$$= -V_\rho(\pi_\nu^\star, (1 - \epsilon_\pi)\widehat{\pi}_\alpha + \epsilon_\pi \pi_\alpha^\star)$$

$$= \min_{\pi_\nu'} -\min_{\pi_\alpha'} V_\rho(\pi_\nu', (1 - \epsilon_\pi)\widehat{\pi}_\alpha + \epsilon_\pi \pi_\alpha')$$

$$= \min_{\pi_\nu'} \mathrm{Expl}(\pi_\nu').$$

$\square$

Therefore, the solution concept NE guarantees the most robust policy for the victim. Indeed, it is stronger than the goal of the most robust policy since by following exactly the same procedure, one can verify NE also guarantees the most robust policy for the *attacker*. However, as we remarked previously, we only care about the robustness of the victim.

## C  MOTIVATION AND EXAMPLES OF TIMESCALE SEPARATION

To motivate the necessity of timescale separation in adversarial training for robust RL policy, we revisit some known issues of naive single-timescale methods including Gradient Descent Ascent (GDA) and iterative best response (IBR) with both simultaneous and alternating update using a simple normal-form game Rock-Paper-Scissor, which corresponds to our defense problem with a single state and $\epsilon_\pi = 1$.

**Example C.1.** *The zero-sum game Rock-Paper-Scissor includes two players with the same action space $\mathcal{A} = \{Rock, Paper, Scissor\}$. The payoff matrix $\mathbf{P}$ is given as $\mathbf{P} = \begin{bmatrix} 0 & 1 & -1 \\ -1 & 0 & 1 \\ 1 & -1 & 0 \end{bmatrix}$ for the row player. The row player has the mixed strategy $x \in \mathcal{X} = \Delta(\mathcal{A})$, where $x_i$ represents the probability of choosing $i^{th}$ action. The column player holds a similar mixed strategy $y \in \mathcal{Y}$. The corresponding payoff is given by $V(x, y) = x^\top \mathbf{P} y$. Our objective is given by $\max_{x \in \mathcal{X}} \min_{y \in \mathcal{Y}} x^\top \mathbf{P} y$. Note here $\epsilon_\pi = 1$.*

Formally, we compare the following 5 methods; the first 4 methods have only a single timescale while the last GAMin method highlights timescale separation, where one player takes a gradient step, and then another one takes the best response.

- Simultaneous gradient descent ascent (SGDA): $y_{t+1} = \mathcal{P}_\mathcal{Y}(y_t - \eta \mathbf{P}^\top x_t)$, $x_{t+1} = \mathcal{P}_\mathcal{X}(x_t + \eta \mathbf{P} y_t)$.
- Alternate gradient descent ascent (AGDA): $y_{t+1} = \mathcal{P}_\mathcal{Y}(y_t - \eta \mathbf{P}^\top x_t)$, $x_{t+1} = \mathcal{P}_\mathcal{X}(x_t + \eta \mathbf{P} y_{t+1})$
- Simultaneous iterative best response (SIBR): $y_{t+1} = \arg\min_{y \in \mathcal{Y}} x_t^\top \mathbf{P} y$, $x_{t+1} = \arg\min_{x \in \mathcal{X}} x^\top \mathbf{P} y_t$
- Alternate iterative best response (AIBR): $y_{t+1} = \arg\min_{y \in \mathcal{Y}} x_t^\top \mathbf{P} y$, $x_{t+1} = \arg\min_{x \in \mathcal{X}} x^\top \mathbf{P} y_{t+1}$.
- (With timescale separation) Gradient ascent with min oracle (GAMin): $y_{t+1} = \arg\min_{y \in \mathcal{Y}} x_t^\top \mathbf{P} y$, $x_{t+1} = \mathcal{P}_\mathcal{X}(x_t + \eta \mathbf{P} y_{t+1})$.

We show the exploitability of the $x$ player during the learning process in Figure 5. It is clear that the first four single-timescale training methods (SGDA, AGDA, SIBR, AIBR) fail to achieve low exploitability, while only GAMin achieves the near-optimal exploitability 0.

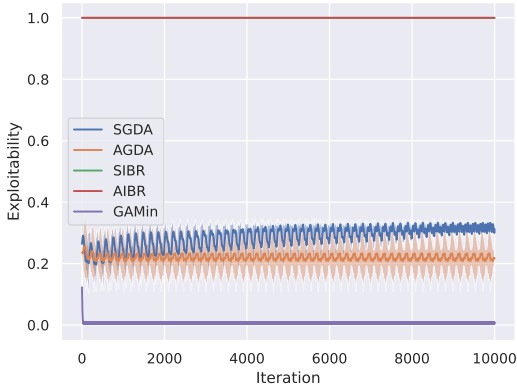

Figure 5: Exploitability test on Rock-Paper-Scissor. Note that the green line is overlapped with the red.

# D FULL PROOF

## D.1 PROOF OF PROPOSITION 3.1 AND 3.2

*Proof.* For simplicity, we shall prove Proposition 3.2 first, and then prove Proposition 3.1.

Let us first review the following facts for any joint policy $\boldsymbol{\pi} = (\pi_\nu, \pi_\alpha)$, $\boldsymbol{\pi}' = (\pi_\nu, \pi'_\alpha)$ such that $D_{\text{TV}}^{\max}(\pi_\alpha||\pi'_\alpha) \le \epsilon_\pi$ and the transition matrix $P_{\boldsymbol{\pi}}$, where $P_{\boldsymbol{\pi}}(s', s) = \sum_{\boldsymbol{a}} \boldsymbol{\pi}(\boldsymbol{a}|s)P(s'|s, \boldsymbol{a})$. In the following proof, we use $P_{\boldsymbol{\pi}}(i, j)$ to denote $P_{\boldsymbol{\pi}}(s_i, s_j)$.

- $d_\rho^{\boldsymbol{\pi}} = (1-\gamma)(I - \gamma P_{\boldsymbol{\pi}})^{-1}\rho$.
- $||P_{\boldsymbol{\pi}}||_1 = 1$ and $||(I - \gamma P_{\boldsymbol{\pi}})^{-1}||_1 \le \frac{1}{1-\gamma}$.
- $||P_{\boldsymbol{\pi}} - P_{\boldsymbol{\pi}'}||_1 \le 2\epsilon_\pi$

According to Definition 2.1, one can verify that $d_\rho^{\boldsymbol{\pi}}$ satisfies that:

$$d_\rho^{\boldsymbol{\pi}} = (1-\gamma)\rho + \gamma P_{\boldsymbol{\pi}} d_\rho^{\boldsymbol{\pi}},$$

which gives the solution $d_\rho^{\boldsymbol{\pi}} = (1-\gamma)(I - \gamma P_{\boldsymbol{\pi}})^{-1}\rho$.

For $P_{\boldsymbol{\pi}}$:

$$||P_{\boldsymbol{\pi}}||_1 = \max_j \sum_i |P_{\boldsymbol{\pi}}(i, j)|$$

$$= \max_j \sum_i \sum_{\boldsymbol{a}} \boldsymbol{\pi}(\boldsymbol{a}|s_j)P(s_i|s_j, \boldsymbol{a})$$

$$= \max_j \sum_{\boldsymbol{a}} \boldsymbol{\pi}(\boldsymbol{a}|s_j) \sum_i P(s_i|s_j, \boldsymbol{a})$$

$$= 1$$

For $||(I - \gamma P_{\boldsymbol{\pi}})^{-1}||_1$:

$$||(I - \gamma P_{\boldsymbol{\pi}})^{-1}||_1 = ||\sum_{k=0}^{\infty}(\gamma P_{\boldsymbol{\pi}})^k||_1 \le \sum_{k=1}^{\infty}||(\gamma P_{\boldsymbol{\pi}})^k||_1 \le \sum_{k=1}^{\infty}\gamma^k||P_{\boldsymbol{\pi}}||_1^k = \frac{1}{1-\gamma}.$$

For $||P_{\boldsymbol{\pi}} - P_{\boldsymbol{\pi}'}||_1$:

To begin with, since $\pi_\nu$ remains unchanged in our proof, let us abuse the notation a little bit and define the marginalized transition $P(s'|s, a_\alpha) = \sum_{a_\nu} \pi_\nu(a_\nu|s)P(s'|s, a_\nu, a_\alpha)$. We have

$$||P_{\boldsymbol{\pi}} - P_{\boldsymbol{\pi}'}||_1 = \max_j \sum_i |P_{\boldsymbol{\pi}}(i, j) - P_{\boldsymbol{\pi}'}(i, j)|$$

$$= \max_j \sum_i |\sum_{a_\alpha} (\pi_\alpha(a_\alpha|s_j) - \pi'_\alpha(a_\alpha|s_j) \sum_{a_\nu} \pi_\nu(a_\nu|s_j)P(s_i|s_j, a_\nu, a_\alpha)|$$

$$= \max_j \sum_i |\sum_{a_\alpha} (\pi_\alpha(a_\alpha|s_j) - \pi'_\alpha(a_\alpha|s_j))P(s_i|s_j, a_\alpha)|.$$

Now fix any index $j$, define $\boldsymbol{m}_i^\top = (P(s_i|s_j, a_\alpha^k))_{k=1}^{|\mathcal{A}_\alpha|}$, $\boldsymbol{M}^\top = (\boldsymbol{m}_1, \cdots, \boldsymbol{m}_{|\mathcal{S}|})$, and $\boldsymbol{n}^\top = (\pi_\alpha(a_\alpha^k|s_j) - \pi'_\alpha(a_\alpha^k|s_j))_{k=1}^{|\mathcal{A}_\alpha|}$. Then the following holds

$$\sum_i |\sum_{a_\alpha} (\pi_\alpha(a_\alpha|s_j) - \pi'_\alpha(a_\alpha|s_j))P(s_i|s_j, a_\alpha)| = \sum_i |\boldsymbol{m}_i^\top \boldsymbol{n}| = ||\boldsymbol{Mn}||_1 \le ||\boldsymbol{M}||_1 ||\boldsymbol{n}||_1 = 2\epsilon_\pi ||\boldsymbol{M}||_1.$$

According to the definition of $\boldsymbol{M}$, it is easy to check

$$||\boldsymbol{M}||_1 = \max_k \sum_i |P(s_i|s_j, a_\alpha^k)| = 1.$$

Therefore, we conclude that for any fixed index $j$, we have

$$\sum_i |\sum_{a_\alpha} (\pi_\alpha(a_\alpha|s_j) - \pi'_\alpha(a_\alpha|s_j))P(s_i|s_j, a_\alpha)| \le 2\epsilon_\pi,$$

which proves $||P_{\boldsymbol{\pi}} - P_{\boldsymbol{\pi}'}||_1 \le 2\epsilon_\pi$.

Now we are ready to prove Proposition 3.2.

$$||d_\rho^{\boldsymbol{\pi}} - d_\rho^{\boldsymbol{\pi}'}||_1 = ||(1 - \gamma)(I - \gamma P_{\boldsymbol{\pi}})^{-1}\rho - (1 - \gamma)(I - \gamma P_{\boldsymbol{\pi}'})^{-1}\rho||_1$$

$$\le (1 - \gamma)||(I - \gamma P_{\boldsymbol{\pi}})^{-1} - (I - \gamma P_{\boldsymbol{\pi}'})^{-1}||_1 ||\rho||_1$$

$$\le (1 - \gamma)||(I - \gamma P_{\boldsymbol{\pi}'})^{-1}||_1 ||\gamma(P_{\boldsymbol{\pi}} - P_{\boldsymbol{\pi}'})||_1 ||(I - \gamma P_{\boldsymbol{\pi}})^{-1}||_1 ||\rho||_1$$

$$\le \frac{2\epsilon_\pi \gamma}{1 - \gamma}.$$

Now we can use Proposition 3.2 to prove Proposition 3.1. To begin with, it is easy to verify that the following holds

$$V_\rho(\boldsymbol{\pi}) = \sum_s d_\rho^{\boldsymbol{\pi}}(s) \sum_a \boldsymbol{\pi}(\boldsymbol{a}|s)r(s, \boldsymbol{a}).$$

Now let us define the marginalized reward $r_{\boldsymbol{\pi}}(s) = \sum_a \boldsymbol{\pi}(\boldsymbol{a}|s)r(s, \boldsymbol{a})$, and further define the vector notation $r_{\boldsymbol{\pi}}^\top = (r_{\boldsymbol{\pi}}(s^k))_{k=1}^{|\mathcal{S}|}$. Then for the difference of the value function, it holds that

$$|V_\rho(\boldsymbol{\pi}) - V_\rho(\boldsymbol{\pi}')| = \frac{1}{1 - \gamma}|\langle d_\rho^{\boldsymbol{\pi}}, r_{\boldsymbol{\pi}}\rangle - \langle d_\rho^{\boldsymbol{\pi}'}, r_{\boldsymbol{\pi}'}\rangle|$$

$$= \frac{1}{1 - \gamma}|\langle d_\rho^{\boldsymbol{\pi}}, r_{\boldsymbol{\pi}}\rangle - \langle d_\rho^{\boldsymbol{\pi}}, r_{\boldsymbol{\pi}'}\rangle + \langle d_\rho^{\boldsymbol{\pi}}, r_{\boldsymbol{\pi}'}\rangle - \langle d_\rho^{\boldsymbol{\pi}'}, r_{\boldsymbol{\pi}'}\rangle|$$

$$\le \frac{1}{1 - \gamma}(|\langle d_\rho^{\boldsymbol{\pi}}, r_{\boldsymbol{\pi}}\rangle - \langle d_\rho^{\boldsymbol{\pi}}, r_{\boldsymbol{\pi}'}\rangle| + |\langle d_\rho^{\boldsymbol{\pi}}, r_{\boldsymbol{\pi}'}\rangle - \langle d_\rho^{\boldsymbol{\pi}'}, r_{\boldsymbol{\pi}'}\rangle|)$$

$$\le \frac{1}{1 - \gamma}(||d_\rho^{\boldsymbol{\pi}}||_1 ||r_{\boldsymbol{\pi}} - r_{\boldsymbol{\pi}'}||_\infty + ||d_\rho^{\boldsymbol{\pi}} - d_\rho^{\boldsymbol{\pi}'}||_1 ||r_{\boldsymbol{\pi}'}||_\infty)$$

$$\le \frac{1}{1 - \gamma}(2\epsilon_\pi + \frac{2\epsilon_\pi \gamma}{1 - \gamma})$$

$$\le \frac{2\epsilon_\pi}{(1 - \gamma)^2}.$$

$\square$

## D.2 PROOF OF PROPOSITION 3.3

*Proof.* For any fixed $s \in \mathcal{S}$ and $a_\nu \in \mathcal{A}_\nu$, the channel that produces the distribution of next state $s'$ based on the input distribution of $a_\alpha$ is exactly $P(\cdot \mid s, a_\nu, \cdot)$. Therefore, applying the data processing inequality for $f$-divergence, we prove our proposition. $\qquad\square$

## D.3 PROOF OF THEOREM 5.3

In the following discussions, we will use $J_{\epsilon_\pi}(\nu, \alpha)$ and $J_{\epsilon_\pi}(\pi_\nu, \pi_\alpha)$ interchangeably according to Definition 5.1. Once we have a bounded mismatch coefficient in Definition 5.2, we are ready to analyze the properties of the function $J_{\epsilon_\pi}(\nu, \alpha)$ in the following lemma.

**Lemma D.1.** *For any $\nu, \nu' \in \Delta(\mathcal{A}_\nu)^{|\mathcal{S}|}$ and $\alpha, \alpha' \in \Delta(\mathcal{A}_\alpha)^{|\mathcal{S}|}$, the function $J_{\epsilon_\pi}$ satisfies*

- *Lipschitzness*

$$\|\nabla_\nu J_{\epsilon_\pi}(\nu, \alpha)\| \leq \frac{\sqrt{|\mathcal{A}_\nu|}}{(1-\gamma)^2},$$

$$\|\nabla_\alpha J_{\epsilon_\pi}(\nu, \alpha)\| \leq \frac{\epsilon_\pi \sqrt{|\mathcal{A}_\alpha|}}{(1-\gamma)^2}.$$

- *Smoothness*

$$\|\nabla_\nu J_{\epsilon_\pi}(\nu, \alpha) - \nabla_\nu J_{\epsilon_\pi}(\nu', \alpha')\| \leq \frac{2\sqrt{|\mathcal{A}_\nu|}}{(1-\gamma)^3}(\sqrt{|\mathcal{A}_\nu|}\|\nu - \nu'\| + \sqrt{|\mathcal{A}_\alpha|}\|\alpha - \alpha'\|),$$

$$\|\nabla_\alpha J_{\epsilon_\pi}(\nu, \alpha) - \nabla_\alpha J_{\epsilon_\pi}(\nu', \alpha')\| \leq \frac{2\epsilon_\pi \sqrt{|\mathcal{A}_\alpha|}}{(1-\gamma)^3}(\sqrt{|\mathcal{A}_\nu|}\|\nu - \nu'\| + \sqrt{|\mathcal{A}_\alpha|}\|\alpha - \alpha'\|).$$

- *Gradient domination*

$$J_{\epsilon_\pi}(\nu, \alpha) - \min_{\alpha'} J_{\epsilon_\pi}(\nu, \alpha') \leq \frac{C_\mathcal{G}^{\epsilon_\pi}}{1-\gamma} \max_{\bar{\alpha}} \langle \nabla_\alpha J_{\epsilon_\pi}(\nu, \alpha), \alpha - \bar{\alpha} \rangle, \qquad (\text{D.1})$$

$$\max_{\nu'} J_{\epsilon_\pi}(\nu', \alpha) - J_{\epsilon_\pi}(\nu, \alpha) \leq \frac{C_\mathcal{G}^{\epsilon_\pi}}{1-\gamma} \max_{\bar{\nu}} \langle \nabla_\nu J_{\epsilon_\pi}(\nu, \alpha), \bar{\nu} - \nu \rangle. \qquad (\text{D.2})$$

*Proof.* With the definition of the function $J_{\epsilon_\pi}$, we have $J_{\epsilon_\pi}(\nu, \alpha) = V_\rho(\pi_\nu, (1 - \epsilon_\pi)\widehat{\pi}_\alpha + \epsilon_\pi \pi_\alpha)$.

- For Lipschitzness, it is easy to compute of gradient of $J_{\epsilon_\pi}$ with respect to $\nu$ and $\alpha$ using chain rules. Let's denote $\pi_\alpha^{mix} = (1 - \epsilon_\pi)\widehat{\pi}_\alpha + \epsilon_\pi \pi_\alpha$. Then the following holds using chain rules and standard policy gradient expression Zhang et al. (2021b).

$$\frac{\partial J_{\epsilon_\pi}(\nu, \alpha)}{\partial \nu_{s,a_\nu}} = \frac{\partial V_\rho(\pi_\nu, \pi_\alpha^{mix})}{\partial \nu_{s,a}} = \frac{1}{1-\gamma} d_\rho^{\pi_\nu, \pi_\alpha^{mix}}(s) \mathbb{E}_{a_\alpha \sim \pi_\alpha^{mix}}[Q^{\pi_\nu, \pi_\alpha^{mix}}(s, a_\nu, a_\alpha)] \leq \frac{d_\rho^{\pi_\nu, \pi_\alpha^{mix}}(s)}{(1-\gamma)^2},$$

$$\frac{\partial J_{\epsilon_\pi}(\nu, \alpha)}{\partial \alpha_{s,a_\alpha}} = \frac{\partial V_\rho(\pi_\nu, \pi_\alpha^{mix})}{\partial \alpha_{s,a_\alpha}} = \frac{\epsilon_\pi}{1-\gamma} d_\rho^{\pi_\nu, \pi_\alpha^{mix}}(s) \mathbb{E}_{a_\nu \sim \pi_\nu}[Q^{\pi_\nu, \pi_\alpha^{mix}}(s, a_\nu, a_\alpha)] \leq \frac{\epsilon_\pi d_\rho^{\pi_\nu, \pi_\alpha^{mix}}(s)}{(1-\gamma)^2}.$$

  Therefore, we have $\|\nabla_\nu J_{\epsilon_\pi}(\nu, \alpha)\| \leq \frac{\sqrt{|\mathcal{A}_\nu|}}{(1-\gamma)^2}, \|\nabla_\alpha J_{\epsilon_\pi}(\nu, \alpha)\| \leq \frac{\epsilon_\pi \sqrt{|\mathcal{A}_\alpha|}}{(1-\gamma)^2}$.

- For Smoothness, it holds that

$$\|\nabla_\nu J_{\epsilon_\pi}(\nu, \alpha) - \nabla_\nu J_{\epsilon_\pi}(\nu', \alpha')\|$$
$$= \|\nabla_{\pi_\nu} V(\pi_\nu, (1 - \epsilon_\pi)\widehat{\pi}_\alpha + \epsilon_\pi \pi_\alpha) - \nabla_{\pi_\nu} V(\pi'_\nu, (1 - \epsilon_\pi)\widehat{\pi}_\alpha + \epsilon_\pi \pi'_\alpha)\|$$
$$\leq \frac{2\sqrt{|\mathcal{A}_\nu|}}{(1-\gamma)^3}(\sqrt{|\mathcal{A}_\nu|}\|\nu - \nu'\| + \sqrt{|\mathcal{A}_\alpha|}\|\alpha - \alpha'\|).$$

  where the last step comes from Lemma 19 in (Zhang et al., 2021b). Similarly, using chain rules, it holds that

$$\|\nabla_\alpha J_{\epsilon_\pi}(\nu, \alpha) - \nabla_\alpha J_{\epsilon_\pi}(\nu', \alpha')\|$$
$$= \epsilon_\pi \|\nabla_{\pi_\alpha^{mix}} V(\pi_\nu, \pi_\alpha^{mix})\big|_{\pi_\alpha^{mix}=(1-\epsilon_\pi)\widehat{\pi}_\alpha + \epsilon_\pi \pi_\alpha} - \nabla_{\pi_\alpha^{mix}} V(\pi'_\nu, \pi_\alpha^{mix})\big|_{\pi_\alpha^{mix}=(1-\epsilon_\pi)\widehat{\pi}_\alpha + \epsilon_\pi \pi'_\alpha}\|$$
$$\leq \frac{2\epsilon_\pi \sqrt{|\mathcal{A}_\alpha|}}{(1-\gamma)^3}(\sqrt{|\mathcal{A}_\nu|}\|\nu - \nu'\| + \sqrt{|\mathcal{A}_\alpha|}\|\alpha - \alpha'\|).$$

- The most challenging step is establishing the gradient domination properties. We will show that our objective $J_{\epsilon_\pi}(\pi_\nu, \pi_\alpha)$ is indeed a value function of some other Markov games $\widetilde{\mathcal{G}}$ with modified transition and reward compared with the original $\mathcal{G}$, which is defined as for any given $s, a_\nu, a_\alpha, s'$,

$$r_\nu^{mix}(s, a_\nu, a_\alpha) = (1 - \epsilon_\pi) \sum_{a'_\alpha} r_\nu(s, a_\nu, a'_\alpha) \widehat{\pi}_\alpha(a'_\alpha \mid s) + \epsilon_\pi r_\nu(s, a_\nu, a_\alpha),$$

$$P^{mix}(s'|s, a_\nu, a_\alpha) = (1 - \epsilon_\pi) \sum_{a'_\alpha} P(s' \mid s, a_\nu, a'_\alpha) \widehat{\pi}_\alpha(a'_\alpha \mid s) + \epsilon_\pi P(s' \mid s, a_\nu, a_\alpha).$$

For this modified Markov game, we denote the corresponding value function as $\widetilde{V}_\rho(\pi_\nu, \pi_\alpha)$, and the stationary state visitation as $\widetilde{d}_\rho^{\pi_\nu, \pi_\alpha}$. Then it is easy to verify that for any $\pi_\nu \in \Pi_\nu$, $\pi_\alpha \in \Pi_\alpha$

$$J_{\epsilon_\pi}(\pi_\nu, \pi_\alpha) = \widetilde{V}_\rho(\pi_\nu, \pi_\alpha), \tag{D.3}$$

$$d_\rho^{\pi_\nu, (1-\epsilon_\pi)\widehat{\pi}_\alpha + \epsilon_\pi \pi_\alpha} = \widetilde{d}_\rho^{\pi_\nu, \pi_\alpha}. \tag{D.4}$$

Similarly, we define the mismatch coefficient for $\widetilde{\mathcal{G}}$ as follows:

$$C_{\widetilde{\mathcal{G}}} := \max\{ \max_{\pi_\nu \in \Pi_\nu} \min_{\pi_\alpha \in \widetilde{\Pi}_\alpha^\star(\pi_\nu)} \left\| \frac{\widetilde{d}_\rho^{\pi_\nu, \pi_\alpha}}{\rho} \right\|_\infty, \max_{\pi_\alpha \in \Pi_\alpha} \min_{\pi_\nu \in \widetilde{\Pi}_\nu^\star(\pi_\alpha)} \left\| \frac{\widetilde{d}_\rho^{\pi_\nu, \pi_\alpha}}{\rho} \right\|_\infty \},$$

where $\widetilde{\Pi}_\alpha^\star(\pi_\nu) := \arg\min_{\pi_\alpha \in \Pi_\alpha} \widetilde{V}_\rho(\pi_\nu, \pi_\alpha)$, and $\widetilde{\Pi}_\nu^\star(\pi_\alpha) := \arg\max_{\pi_\nu \in \Pi_\nu} \widetilde{V}_\rho(\pi_\nu, \pi_\alpha)$. Next, we use the gradient domination property of the value function of Lemma 3 (Zhang et al., 2021b). It holds that for any $\pi'_\alpha$ and $\pi'_\nu$:

$$\widetilde{V}_\rho(\pi'_\nu, \pi_\alpha) - \widetilde{V}_\rho(\pi_\nu, \pi_\alpha) \le \left\| \frac{\widetilde{d}_\rho^{\pi'_\nu, \pi_\alpha}}{\widetilde{d}_\rho^{\pi_\nu, \pi_\alpha}} \right\|_\infty \max_{\bar{\pi}_\nu} \langle \nabla_{\pi_\nu} \widetilde{V}_\rho(\pi_\nu, \pi_\alpha), \bar{\pi}_\nu - \pi_\nu \rangle,$$

$$\widetilde{V}_\rho(\pi_\nu, \pi_\alpha) - V_\rho(\pi_\nu, \pi'_\alpha) \le \left\| \frac{\widetilde{d}_\rho^{\pi_\nu, \pi'_\alpha}}{\widetilde{d}_\rho^{\pi_\nu, \pi_\alpha}} \right\|_\infty \max_{\bar{\pi}_\alpha} \langle \nabla_{\pi_\alpha} \widetilde{V}_\rho(\pi_\nu, \pi_\alpha), \pi_\alpha - \bar{\pi}_\alpha \rangle.$$

To prove Equation equation D.1, we denote $\pi_\alpha^\star \in \widetilde{\Pi}_\alpha^\star(\pi_\nu)$ to be the policy minimizing the quantity $\left\| \frac{\widetilde{d}_\rho^{\pi_\nu, \pi_\alpha}}{\rho} \right\|_\infty$. Then, we have

$$\begin{aligned}
J_{\epsilon_\pi}(\nu, \alpha) - \min_{\alpha'} J_{\epsilon_\pi}(\nu, \alpha') &= \widetilde{V}_\rho(\pi_\nu, \pi_\alpha) - \min_{\pi'_\alpha} \widetilde{V}_\rho(\pi_\nu, \pi'_\alpha) \\
&= \widetilde{V}_\rho(\pi_\nu, \pi_\alpha) - \widetilde{V}_\rho(\pi_\nu, \pi_\alpha^\star) \\
&\le \left\| \frac{\widetilde{d}_\rho^{\pi_\nu, \pi_\alpha^\star}}{\widetilde{d}_\rho^{\pi_\nu, \pi_\alpha}} \right\|_\infty \max_{\bar{\pi}_\alpha} \langle \nabla_{\pi_\alpha} \widetilde{V}_\rho(\pi_\nu, \pi_\alpha), \bar{\pi}_\alpha - \pi_\alpha \rangle \\
&\le \frac{1}{1 - \gamma} \left\| \frac{\widetilde{d}_\rho^{\pi_\nu, \pi_\alpha^\star}}{\rho} \right\|_\infty \max_{\bar{\pi}_\alpha} \langle \nabla_{\pi_\alpha} \widetilde{V}_\rho(\pi_\nu, \pi_\alpha), \bar{\pi}_\alpha - \pi_\alpha \rangle \\
&\le \frac{1}{1 - \gamma} C_{\widetilde{\mathcal{G}}} \max_{\bar{\pi}_\alpha} \langle \nabla_{\pi_\alpha} \widetilde{V}_\rho(\pi_\nu, \pi_\alpha), \bar{\pi}_\alpha - \pi_\alpha \rangle.
\end{aligned}$$

Note due to Equation equation D.3 and equation D.4, we have $C_{\widetilde{\mathcal{G}}} = C_{\mathcal{G}}^{\epsilon_\pi}$, and $\nabla_{\pi_\alpha} \widetilde{V}_\rho(\pi_\nu, \pi_\alpha) = \nabla_{\pi_\alpha} J_{\epsilon_\pi}(\pi_\nu, \pi_\alpha) = \nabla_\alpha J_{\epsilon_\pi}(\nu, \alpha)$, proving Equation equation D.1.

To prove Equation equation D.2, we denote $\pi_\nu^\star \in \widetilde{\Pi}_\nu^\star(\pi_\alpha)$ to be the policy minimizing the quantity $\left\| \frac{\widetilde{d}_\rho^{\pi_\nu, \pi_\alpha}}{\rho} \right\|_\infty$. Therefore, we have that

$$
\begin{aligned}
\max_{\nu'} J_{\epsilon_\pi}(\nu', \alpha) - J_{\epsilon_\pi}(\nu, \alpha) &= \max_{\pi_\nu'} \widetilde{V}_\rho(\pi_\nu', \pi_\alpha) - \widetilde{V}_\rho(\pi_\nu, \pi_\alpha) \\
&= \widetilde{V}_\rho(\pi_\nu^\star, \pi_\alpha) - \widetilde{V}_\rho(\pi_\nu, \pi_\alpha) \\
&\leq \left\| \frac{\widetilde{d}_\rho^{\pi_\nu^\star, \pi_\alpha}}{\widetilde{d}_\rho^{\pi_\nu, \pi_\alpha}} \right\|_\infty \max_{\bar{\pi}_\nu} \langle \nabla_{\pi_\nu} \widetilde{V}_\rho(\pi_\nu, \pi_\alpha), \bar{\pi}_\nu - \pi_\nu \rangle \\
&\leq \frac{1}{1-\gamma} \left\| \frac{\widetilde{d}_\rho^{\pi_\nu^\star, \pi_\alpha}}{\rho} \right\|_\infty \max_{\bar{\pi}_\nu} \langle \nabla_{\pi_\nu} \widetilde{V}_\rho(\pi_\nu, \pi_\alpha), \bar{\pi}_\nu - \pi_\nu \rangle \\
&\leq \frac{1}{1-\gamma} C_{\widetilde{\mathcal{G}}} \max_{\bar{\pi}_\nu} \langle \nabla_{\pi_\nu} \widetilde{V}_\rho(\pi_\nu, \pi_\alpha), \bar{\pi}_\nu - \pi_\nu \rangle.
\end{aligned}
$$

Note due to Equation equation D.3 and equation D.4, we have $C_{\widetilde{\mathcal{G}}} = C_{\mathcal{G}}^{\epsilon_\pi}$, and $\nabla_{\pi_\nu} \widetilde{V}_\rho(\pi_\nu, \pi_\alpha) = \nabla_{\pi_\nu} J_{\epsilon_\pi}(\pi_\nu, \pi_\alpha) = \nabla_\nu J_{\epsilon_\pi}(\nu, \alpha)$, proving Equation equation D.2.

$\square$

Before proving Theorem 5.3, we need the following additional lemmas. Firstly, we define $\phi(\cdot) := \min_\alpha J_{\epsilon_\pi}(\cdot, \alpha)$, and the Moreau envelope for any $\lambda > 0$ as $\phi_\lambda(\nu) := \max_{\nu'} \phi(\nu') - \frac{1}{2\lambda} \|\nu - \nu'\|^2$.

**Lemma D.2** (Theorem 31 (Jin et al., 2020)). *Suppose the function $J_{\epsilon_\pi}$ is $l$-smooth and $L$-Lipschitz. Then the output of Algorithm 1 with step size $\eta^t = \frac{1}{\sqrt{T+1}}$ satisfies*

$$
\frac{1}{T} \sum_{t=1}^T \|\nabla \phi_{1/2l}(\nu^t)\|^2 \leq 2 \cdot \frac{\max_\nu \phi(\nu) - \phi(\nu^0) + lL^2}{\sqrt{T+1}}.
$$

**Lemma D.3** (Lemma 12 (Daskalakis et al., 2020)). *Suppose $J_{\epsilon_\pi}$ is $l$-smooth and $L$-Lipschitz. If there is some $u_\nu$ such that for any $\nu$, $\alpha$*

$$
\max_{\nu^\star} J_{\epsilon_\pi}(\nu^\star, \alpha) - J_{\epsilon_\pi}(\nu, \alpha) \leq u_\nu \max_{\bar{\nu}} \langle \nabla_\nu J_{\epsilon_\pi}(\nu, \alpha), \bar{\nu} - \nu \rangle,
$$

*then it holds that*

$$
\max_{\nu^\star} \phi(\nu^\star) - \phi(\nu) \leq (u_\nu + \frac{L}{2l}) \|\nabla \phi_{1/2l}(\nu)\|^2.
$$

**Lemma D.4** (Theorem 2a (Daskalakis et al., 2020)). *Suppose $J_{\epsilon_\pi}$ is $l$-smooth and $L$-Lipschitz. If there is some $u_\nu$, $u_\alpha$ such that for any $\nu$, $\alpha$*

$$
\max_{\nu^\star} J_{\epsilon_\pi}(\nu^\star, \alpha) - J_{\epsilon_\pi}(\nu, \alpha) \leq u_\nu \max_{\bar{\nu}} \langle \nabla_\nu J_{\epsilon_\pi}(\nu, \alpha), \bar{\nu} - \nu \rangle,
$$
$$
J_{\epsilon_\pi}(\nu, \alpha) - \min_{\alpha^\star} J_{\epsilon_\pi}(\nu, \alpha^\star) \leq u_\alpha \max_{\bar{\alpha}} \langle \nabla_\alpha J_{\epsilon_\pi}(\nu, \alpha), \alpha - \bar{\nu} \rangle.
$$

*Then it holds that for $\eta_\alpha^t = \eta_\alpha = \Theta\left( \frac{\delta^4 u_\alpha^2}{l^3 L^2 (L/l+1)^2} \right)$, and $\eta_\nu^t = \eta_\nu = \Theta\left( \min\{ \frac{\delta^8 u_\alpha^4}{l^5 (L/l+1)^4 L^4}, \frac{\delta^2}{lL^2} \} \right)$, we have that*

$$
\frac{1}{T} \sum_{t=1}^T \max_{\nu^\star} \phi(\nu^\star) - \phi(\nu^t) \leq \left( u_\nu + \frac{L}{2l} \right) \delta,
$$

*for $T \geq \Omega\left( \frac{(D_{\Pi_\nu} + D_{\Pi_\alpha})L}{\delta^2 \eta_\nu} \right)$ iterations, where $D_{\Pi_\nu}$ and $D_{\Pi_\alpha}$ is the radius of the set $\Pi_\nu$, and $\Pi_\alpha$.*

Now we are ready to prove Theorem 5.3.

*Proof.* We start from proving the convergence of Algorithm 1. By substituting the parameters into Lemma D.2, we notice that the parameters satisfy that $L \le \frac{|\mathcal{A}_\nu| + |\mathcal{A}_\alpha|}{(1-\gamma)^2}, l \le \frac{2(|\mathcal{A}_\nu| + |\mathcal{A}_\alpha|)}{(1-\gamma)^3}, u_\nu = \frac{C_\mathcal{G}^{\epsilon_\pi}}{1-\gamma}$ and $\phi(\nu) \le \frac{1}{1-\gamma}$. Therefore, we get

$$\frac{1}{T} \sum_{t=1}^{T} \|\nabla \phi_{1/2l}(\nu^t)\|^2 \le 2 \cdot \frac{\max_\nu \phi(\nu) - \phi(\nu^0) + lL^2}{\sqrt{T+1}} \le \frac{2}{(1-\gamma)\sqrt{T+1}} + \frac{4(|\mathcal{A}_\nu| + |\mathcal{A}_\alpha|)^2}{(1-\gamma)^7 \sqrt{T+1}}.$$

Now we use Lemma D.3, and conclude that

$$\frac{1}{T} \sum_{t=1}^{T} \max_{\nu^\star} \phi(\nu^\star) - \phi(\nu^t) \le \frac{1}{\sqrt{T+1}} \operatorname{poly}(\frac{1}{1-\gamma}, |\mathcal{S}|, |\mathcal{A}_\nu|, |\mathcal{A}_\alpha|, C_\mathcal{G}^{\epsilon_\pi}).$$

Therefore, for Algorithm 1 to achieve accuracy $\delta$, combined with the fact that $\phi(\cdot) = -\operatorname{Expl}(\cdot)$, one needs $T = \frac{1}{\delta^2} \operatorname{poly}(\frac{1}{1-\gamma}, |\mathcal{S}|, |\mathcal{A}_\alpha|, |\mathcal{A}_\nu|, C_\mathcal{G}^{\epsilon_\pi})$ total number of iterations, concluding the proof for Algorithm 1.

The proof for Algorithm 2 follows similar steps. We notice the instantiation of the parameters $L \le \frac{|\mathcal{A}_\nu| + |\mathcal{A}_\alpha|}{(1-\gamma)^2}, l \le \frac{2(|\mathcal{A}_\nu| + |\mathcal{A}_\alpha|)}{(1-\gamma)^3}, u_\nu, u_\alpha \le \frac{C_\mathcal{G}^{\epsilon_\pi}}{1-\gamma}, D_{\Pi_\nu}, D_{\Pi_\alpha} \le \sqrt{|\mathcal{S}|}$. Bt substituting the parameters into Lemma D.4 and combing it with the fact that $\phi(\cdot) = -\operatorname{Expl}(\cdot)$, we set total number of iterations $T = \operatorname{poly}(\frac{1}{\delta}, C_\mathcal{G}^{\epsilon_\pi}, |\mathcal{S}|, |\mathcal{A}_\alpha|, |\mathcal{A}_\nu|, \frac{1}{1-\gamma})$ to achieve accuracy $\delta$, concluding the proof for Algorithm 2. $\square$

# E    EXPERIMENTAL DETAILS

## E.1    INTRODUCTION TO THE ENVIRONMENT

Kuhn Poker is a popular research game, which is extensive-form and zero-sum with discrete observation and action space. There exists an efficient min oracle with game tree search. For Robosumo competition, both agents are multi-leg robots and observe the position, velocity, and contact forces of joints in their body, and the position of their opponent's joints, which is much more challenging due to the high dimensional observation and action space.

## E.2    FEATURE IMPORTANCE IN ROBOSUMO ENVIRONMENTS

We show the important features for the Robosumo environment in Table 1, 2, 3.

| Original | Local | Global |
|---|---|---|
| 38 | 32 | 31 |
| 32 | 43 | 39 |
| 33 | 38 | 29 |
| 43 | 37 | 43 |
| 41 | 31 | 41 |
| 29 | 35 | 34 |
| 35 | 42 | 42 |
| 39 | 41 | 35 |
| 42 | 40 | 37 |
| 26 | 28 | 44 |

Table 1: Feature Importance (Subset) in Robosumo Spider vs Spider. "Original": Index of the most important features of the victim's observation while playing against normal opponents. "Local": Index of the most important features of the victim's observation while playing against the attacker trained via our constrained attack method. "Global": Index of the most important features of the victim's observation while playing against the attacker trained via the unconstrained attack method as done in (Gleave et al., 2019)

.

| Original | Local | Global |
|---|---|---|
| 25 | 25 | 23 |
| 145 | 26 | 146 |
| 26 | 145 | 145 |
| 146 | 24 | 25 |
| 24 | 22 | 22 |
| 142 | 146 | 21 |
| 144 | 23 | 24 |
| 22 | 143 | 144 |
| 143 | 27 | 142 |
| 23 | 142 | 143 |

Table 2: Feature Importance (Subset) in Robosumo Ant vs Ant. "Original": Index of the most important features of the victim's observation while playing against normal opponents. "Local": Index of the most important features of the victim's observation while playing against the attacker trained via our constrained attack method. "Global": Index of the most important features of the victim's observation while playing against the attacker trained via the unconstrained attack method as done in (Gleave et al., 2019)

.

| Original | Local | Global |
|----------|-------|--------|
| 193 | 27 | 33 |
| 197 | 35 | 25 |
| 25 | 25 | 29 |
| 35 | 197 | 27 |
| 27 | 193 | 35 |
| 29 | 189 | 31 |
| 191 | 199 | 36 |
| 199 | 34 | 34 |
| 190 | 191 | 28 |
| 36 | 36 | 26 |

Table 3: Feature Importance (Subset) in Robosumo Bug vs Bug. "Original": Index of the most important features of the victim's observation while playing against normal opponents. "Local": Index of the most important features of the victim's observation while playing against the attacker trained via our constrained attack method. "Global": Index of the most important features of the victim's observation while playing against the attacker trained via the unconstrained attack method as done in (Gleave et al., 2019)

.

### E.3 IMPLEMENTATION DETAILS.

**Kuhn Poker.** For implementing our adversarial training approach, we adopt both advanced A2C and RPG policy gradient methods. The policy is parameterized by an MLP with a hidden layer size of $128$. We use a batch size of $16$ and SGD as the optimizer.

**Robosumo competition.** We use PPO as our base policy optimization algorithm. We use a clip range of $0.2$ and $0.95$ for GAE. The number of hidden units for parameterization of the policy is $64$. For all adversarial training algorithms with both a single timescale and two timescales and the baseline algorithm self-play and fictitious-play, we train by $3e7$ timesteps with a batch size of $64$. For attack, we train for $2e7$ timesteps with a batch size of $2048$. We use Adam as the optimizer.

### E.4 ADDITIONAL EXPERIMENTAL RESULTS

**Trade-off between stealthiness and winning rate.** We have mentioned that there is a trade-off between stealthiness and the winning rate. To quantitatively show the trade-off, we show the comparison of our constrained attack and unconstrained one in Figure 6, where $\epsilon_\pi = \lambda$ in the figures.

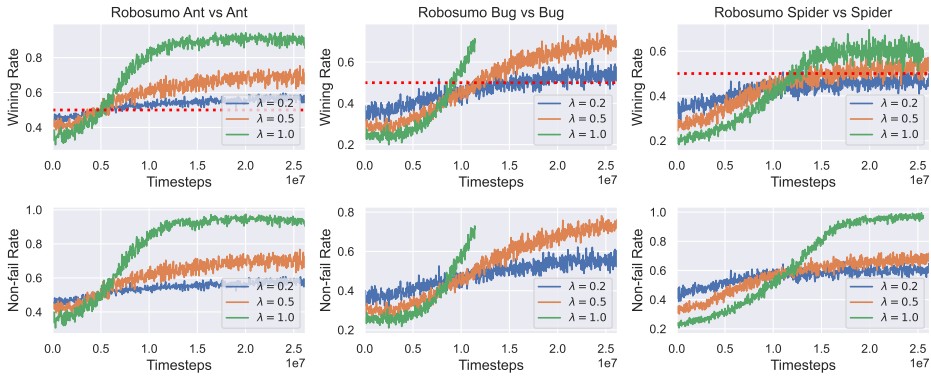

Figure 6: Winning rate and Non-fail rate of different attack methods.

**Additional results on the state-distribution shifts.** To quantify the state-distribution shift induced due to the unconstrained (global) and constraint (local) attacks, in addition to the Wasserstein

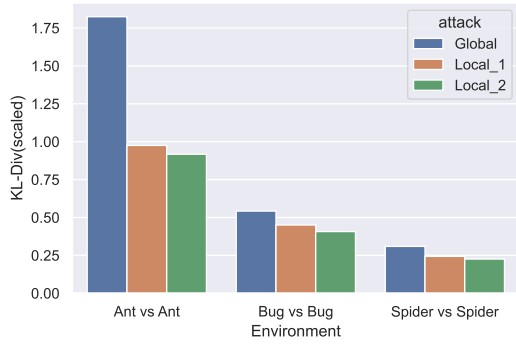

(a) KL-Divergence(Scaled)

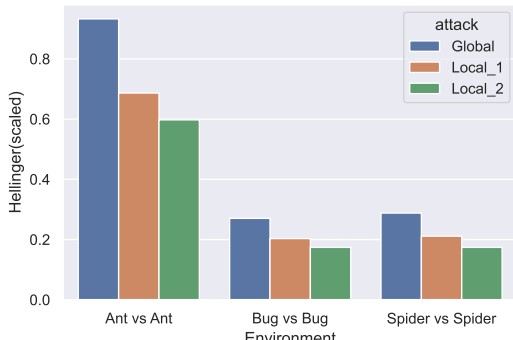

(b) Hellinger Distance(Scaled)

Figure 7: This figure compares the state-distribution shift w.r.t KL-Divergence(Scaled) and Hellinger Distance(scaled) incurred due to the Global ($\epsilon_\pi = 1$), Local-1 ($\epsilon_\pi = 0.7$), Local-2($\epsilon_\pi = 0.3$) attacks in 3 Robosumo environments. The plot clearly demonstrates the Stealthiness of constraint (local) attacks in preserving the state-distribution shifts and achieving stealthiness.

distance in Figure 3, we also compute the KL-Divergence and Hellinger distance between the state-distribution of the unattacked policy with the global and local attacks respectively. We observe that the state-distribution shift incurred is much lesser in local attacks than global, as also can be seen in Figure 1,3, highlighting the stealthiness of our attack.

**Mismatch of attack budgets between training time and test time.** We show the attacker's performance when there is a mismatch between the actual attack budget and the budget used to train the robust victim in the following table. It is clear that **even if the defense budget is not correctly specified, the victim policy still shows greatly improved robustness**, although a bit worse than the case when the defense budget is correctly specified.

| Attacker score | Attack budget 0.3 | Attack budget 0.7 | Attack budget 1.0 |
|---|---|---|---|
| Defense budget 0.3 | **0.38** | 0.50 | 0.71 |
| Defense budget 0.7 | 0.41 | **0.43** | 0.53 |
| Defense budget 1.0 | 0.44 | 0.45 | **0.50** |
| No defense | 0.59 | 0.78 | 0.90 |

Table 4: Attacker's score with different attack budgets against robust victim policies with different defense budgets and without defenses

