# OpenReview forum: "Rethinking Adversarial Policies: A Generalized Attack Formulation and Provable Defense in RL"
_ICLR.cc/2024/Conference — ICLR 2024 poster_

### Official Review · Reviewer_gybG · 2023-10-31

**Soundness:** 3 good
**Presentation:** 2 fair
**Contribution:** 3 good
**Rating:** 6
**Confidence:** 4

**Summary:**

The paper considers a two-player Markov game where an attacker controls one player to minimize the performance of the victim player. Thus, the game is essentially zero-sum. The setting has been considered in some recent work. The paper considers an extension where a player is partially controlled by the attacker and follows the original policy with a certain probability and the malicious policy otherwise. The paper derives some analytic results regarding the effects of attack budget on attack detectability. It then formulates the victim's problem of minimizing the one-side exploitability, proposes two algorithms, and establishes their convergence.

**Strengths:**

Previous work on adversarial policies has mainly focused on unconstrained attacks, which can be easily detected. Thus, it makes sense to consider a constrained attacker. The paper shows that the approach leads to stealthier attacks with significantly lower state distribution shifts.

The paper establishes the convergence of adversarial training with an inner oracle and two-timescale adversarial training for two-player zero-sum Markov games when the attacker is constrained by extending previous results on the unconstrained case, which may find applications in other settings where asymmetric Markov games with constrained followers can be applied

**Weaknesses:**

It seems that minimizing the so-called one-side exploitability (definition 4.1) is the same as finding the Stackelberg equilibrium, a well-studied concept in game theory and multi-agent RL. Both Algorithm 1 and Algorithm 2 follow the general framework for finding the Stackelberg equilibrium in deep multi-agent RL summarized in Gerstgrasser et al. (2023). The convergence analysis of the two-timescale approach follows Daskalakis et al. (2020) and Zeng et al. (2022). Thus, the technical contribution of the paper seems limited.

The paper assumes that the original policy \hat{\pi}_alpha of the agent partially controlled by the attacker is fixed so that existing algorithmic and analytic frameworks can be applied. However, this is not a reasonable assumption. Since the agent is playing a game with the victim (even without the attacker), it should adapt its (original) policy against the victim's policy rather than using a fixed policy.

In Section 3, since the victim's policy is assumed to be fixed, the problem reduces to a single agent MDP, and Propositions 3.1-3.3 are simple variants of well-known results in approximate dynamic programming. Also, I don't see the purpose of mentioning concepts such as generative modeling and distribution matching.

Matthias Gerstgrasser and David C. Parkes. Oracles & Followers: Stackelberg Equilibria in Deep Multi-Agent Reinforcement Learning. 2023.

**Questions:**

Although I agree that directly perturbing the perceived state of an agent is not always feasible in practice, I don't see why controlling another agent against the victim is easier, as claimed in the introduction. They are just different types of attacks. In the air traffic control example, what is the Markov game between the radar system and the commercial drone?

Although it makes sense to consider a constrained attack model to obtain better stealthiness, the proposed formulation is a bit confusing. In particular, if the agent is only partially controlled by the attacker, there is a three-player game between the victim, the agent partially controlled by the attacker, and the attacker.  In this case, why should the agent adopt a fixed policy without adapting to the victim's policy?

In Figure 1, it says that an unconstrained adversarial policy and a constrained policy with the same winning rate are chosen. How is this achieved? Also, is the same victim policy used in both cases? Intuitively, a smaller epsilon should produce a lower state distribution shift and a stealthier attack, but at the same time, the victim's (best possible) performance should improve, given that the attacker becomes weaker.

---

> ### Author Response · Authors · 2023-11-20
>
> We thank Reviewer gybG for the detailed comment and insightful feedback. We are encouraged the reviewer finds our paper "makes sense to consider a constrained attacker" and "find applications in other settings". We address Reviewer gybG's concerns and questions below:
>
> ---
> ### [1/2] Response to weakness
> > Q1. It seems that minimizing the so-called one-side exploitability (definition 4.1) is the same as finding the Stackelberg equilibrium, a well-studied concept in game theory and multi-agent RL. Both Algorithm 1 and Algorithm 2 follow the general framework for finding the Stackelberg equilibrium in deep multi-agent RL summarized in Gerstgrasser et al. (2023). The convergence analysis of the two-timescale approach follows Daskalakis et al. (2020) and Zeng et al. (2022). Thus, the technical contribution of the paper seems limited.
>
> Firstly, we thank the reviewer for mentioning Gerstgrasser et al. (2023). **We acknowledge it is an important related work contributing to the general framework of multi-agent RL and have included it in our revised paper**. However, we believe it does weaken our novelty as we are targeting a more concrete problem and would like to point out the differences:
>
> - **Stackelberg equilibrium is a more general solution concept, and we have discussed more relevant equilibrium concepts, like Nash equilibrium and max-min equilibrium**. Firstly, the solution to one-side exploitability minimization is indeed a max-min equilibrium for the zero-sum game. We acknowledge that stackelberg equilibrium can be reduced to the max-min equilibrium in zero-sum games. However, we believe such terminology stackelberg equilibrium is often used for more general settings beyond zero-sum games and we used **more relevant and specific solution concepts, like Nash equilibrium and max-min equilibrium instead of stackelberg equilibrium**.
> - **Our Algorithm 2 does not follow Gerstgrasser et al. 2023**. Our prototype Algorithm 1 is indeed similar to the framework of Gerstgrasser et al. 2023. Specifically, the attacker can be treated as the follower and compute the best response first, and then the victim, i.e., the leader will update accordingly. However, in Algorithm 2 both the attacker and the victim are updating its policy **simultaneously and independently**. Therefore, **the roles of the victim and attacker in Algorithm 2 are rather symmetric**, for which we believe it is more appropriate to call it an independent learning paradigm instead of a leader-follower framework. **Meanwhile, for theoretical guarantees, Gerstgrasser et al. 2023 point out that "*leader learning problem can be constructed as a POMDP*", which is a computationally intractable problem, while we can indeed get a polynomial convergence**. This also highlights a significant methodological divergence.
> - **Our analysis does not follow Zeng et al. (2022) since it heavily relies on entropy regularization, while Daskalakis et al. (2020) requires less practical assumptions.** Apart from the more general constraints we considered, since Daskalakis et al. (2020) requires the Markov game to stop at every step with a strictly positive probability, it is less consistent with the empirical applications. Therefore, we consider the more popular infinite-horizon discount reward setting and re-establish the guarantees from scratch, which is indeed non-trivial and can be of independent interests.

---

> > ### Author Response · Authors · 2023-11-20
> > **Response to Reviewer gybG**
> >
> > ---
> > > Q2. The paper assumes that the original policy \hat{\pi}_alpha of the agent partially controlled by the attacker is fixed so that existing algorithmic and analytic frameworks can be applied. However, this is not a reasonable assumption. Since the agent is playing a game with the victim (even without the attacker), it should adapt its (original) policy against the victim's policy rather than using a fixed policy.
> > - **Note that from the perspective of the victim, it is also playing a game with the agent $\alpha$. However, related works [1, 2, 3] having been also focusing on a fixed instead of an adaptive victim.** In other words, the fact agents are playing games are mainly affecting their training process, while at test time, a predetermined policy pair, often an equilibrium, is implemented following training.
> > - **Once the victim agent is assumed to use a fixed policy, it is not necessary for the agent $\alpha$ to deploy an adaptive policy either**. This simplification arises because the dynamics of the interaction are essentially static from the perspective of agent $\alpha$.
> > - **$\hat{\pi}\_\alpha$ being fixed at *TEST TIME* does not contradict to the fact it can be already adapted against the victim policy $\hat{\pi}\_\nu$ by the joint policy learning of $(\hat{\pi}\_\alpha, \hat{\pi}\_\nu)$.** The fact that $\hat{\pi}\_\alpha$ remains fixed during the test phase does not negate its potential adaptation to the victim policy $\hat{\pi}\_\nu$ during joint policy training. In essence, before any attack occurs, $(\hat{\pi}\_\alpha, \hat{\pi}\_\nu)$ may have already undergone training within a game setting, allowing them to adjust to each other's strategies. Our framework is applicable to any policy $\hat{\pi}\_\alpha$, including one that has been co-trained with $\hat{\pi}\_\nu$.
> > ---
> > > Q3. In Section 3, since the victim's policy is assumed to be fixed, the problem reduces to a single agent MDP, and Propositions 3.1-3.3 are simple variants of well-known results in approximate dynamic programming. Also, I don't see the purpose of mentioning concepts such as generative modeling and distribution matching.
> >
> > **The reason why we mention generative modeling and distribution matching is to justify why discrepancy of distribution of states is a reasonable metric for stealthiness.** **The reason why we mention generative modeling and distribution matching is to justify why discrepancy of distribution of states is a reasonable metric for stealthiness.** Note that we are not aware of existing works on characterizing how stealthy an adversarial policy is. Therefore, inspired by generative modelling whose objective is to produce visually natural/similar images by matching distributions, we believe smaller discrepancies on the distributions indicates more stealthiness agent behaviors.
> >
> > ---
> > ### [2/2] Response to questions
> > > Q4.1 Although I agree that directly perturbing the perceived state of an agent is not always feasible in practice, I don't see why controlling another agent against the victim is easier, as claimed in the introduction. They are just different types of attacks.
> >
> > **We have never claimed that controlling another agent against the victim is EASIER. In agreement with the reviewer, we recognize that these are distinct types of attacks, each with its own significance.** Our assertion is that when state-perturbation attacks may not be viable in certain scenarios, an adversarial policy could serve as an alternative means of implementing attacks. We shall make it more clear in our revision.
> > > Q4.2 In the air traffic control example, what is the Markov game between the radar system and the commercial drone?
> >
> > In the air traffic control example, the primary interaction between the radar system (representing aircraft ground handling) and the commercial drone is centered on ensuring the safety of airplanes during takeoff and landing. For instance, a commercial drone might be employed to verify whether ground conditions are favorable for takeoff. Under normal circumstances, the commercial drone and the radar system operate without interference. However, if compromised, a malicious commercial drone could be manipulated to enter the radar's detection range intentionally, creating a false impression of active air traffic. This deception could lead to unnecessary delays in airplane takeoffs.

---

> ### Author Response · Authors · 2023-11-20
> **Response to Reviewer gybG**
>
> ---
> > Q5. Although it makes sense to consider a constrained attack model to obtain better stealthiness, the proposed formulation is a bit confusing. In particular, if the agent is only partially controlled by the attacker, there is a three-player game between the victim, the agent partially controlled by the attacker, and the attacker. In this case, why should the agent adopt a fixed policy without adapting to the victim's policy?
>
> In line with our answer to Question 2:
>
> - **Fixed policy assumption at test time**: It is a plausible assumption that a fixed policy is employed during the test phase. This is consistent with the premise adopted in related works [1, 2, 3] where the victim agent is also presumed to utilize a fixed policy at test time. Consequently, it becomes unnecessary for agent $\alpha$ to implement an adaptive strategy in response to the victim's actions, given the static nature of the interaction during testing.
> - **Pre-attack training and framework applicability**: Prior to any attack, the policy pair $(\hat{\pi}\_\alpha, \hat{\pi}\_\nu)$ undergoes joint training within game scenarios, enabling them to adapt to each other. Importantly, our framework is versatile and applicable to any policy $\hat{\pi}\_\alpha$, including those that have been co-trained with $\hat{\pi}\_\nu$. This adaptability enhances the robustness and relevance of our approach across various scenarios.
> ---
> > Q6. In Figure 1, it says that an unconstrained adversarial policy and a constrained policy with the same winning rate are chosen. How is this achieved? Also, is the same victim policy used in both cases? Intuitively, a smaller epsilon should produce a lower state distribution shift and a stealthier attack, but at the same time, the victim's (best possible) performance should improve, given that the attacker becomes weaker.
> - Firstly, the same victim policy used in both cases.
> - **Secondly, it is true that varying attack budgets will not yield the same rewards or win rates.** In **Appendix E4**, we shows the trade-off: reduced budgets boost stealth but might limit the potential of achieving higher rewards.
> - **Thirdly, the selected policies for the unconstrained attack do NOT necessarily reflect the algorithms' final state**. It's important to clarify that the policies selected for comparison in the unconstrained attack scenario do not necessarily represent the ultimate outcome of the algorithms. For instance, if an adversarial policy developed through our method attains a win rate of 0.7, we juxtapose it with an unconstrained adversarial policy also having a win rate of 0.7. However, such an unconstrained policy may not reflect the final convergence point of the baseline algorithm. The key takeaway is that when an attacker prioritizes stealthiness while targeting a specific win rate, our constrained attack methodology demonstrates superior stealth performance.
>
>
> ---
> We greatly appreciate Reviewer gybG's valuable feedback and constructive suggestions. We are happy to answer any further questions.
>
>
> [1] Adam Gleave, Michael Dennis, Cody Wild, Neel Kant, Sergey Levine, and Stuart Russell. Adver- sarial policies: Attacking deep reinforcement learning. arXiv preprint arXiv:1905.10615, 2019.
>
> [2] Xian Wu, Wenbo Guo, Hua Wei, and Xinyu Xing. Adversarial policy training against deep rein- forcement learning. In 30th {USENIX} Security Symposium ({USENIX} Security 21), 2021b.
>
> [3] Wenbo Guo, Xian Wu, Sui Huang, and Xinyu Xing. Adversarial policy learning in two-player competitive games. In International Conference on Machine Learning, pp. 3910–3919. PMLR, 2021.
>
> Paper2121 Authors

---

> ### Comment · Reviewer_gybG · 2023-11-22
>
> Thanks for the detailed response. Although it certainly makes sense to pretrain $(\hat{\pi}_\alpha,\hat{\pi}_v)$ before attacks occur, I wonder when adversarial training (Algorithm 1) is applied. When the robust victim's policy is trained using adversarial training, shouldn't $\hat{\pi}_a$ be jointly trained as well?

---

> ### Author Response · Authors · 2023-11-22
> **Thanks for your reply!**
>
> > Q. Thanks for the detailed response. Although it certainly makes sense to pretrain $(\hat{\pi}\_\alpha,\hat{\pi}\_v)$ before attacks occur, I wonder when adversarial training (Algorithm 1) is applied. When the robust victim's policy is trained using adversarial training, shouldn't $\hat{\pi}_v$ be jointly trained as well?
>
> Thanks a lot for your question! We find your point on the joint adversarial training of $\hat{\pi}\_\nu$ and $\hat{\pi}\_\alpha$ against $\pi_\alpha$ both interesting and complex. There are indeed various scenarios in terms of the target of the attacker $\pi\_\alpha$. Our approach, while specific, still offers meaningful insights into this broader challenge, alongside contributions from works like [1].
>
> - **[1] tackles the scenario where $\pi_\alpha$ is attacking agent $\alpha$, i.e., $\hat{\pi}\_\alpha$, while keeping $\hat{\pi}\_\nu$ fixed.** This is akin to a single-agent setup, aligning with the situation where $\hat{\pi}\_\nu$ remains unchanged. In such cases, the focus is on preparing $\hat{\pi}\_\alpha$ for potential attacks.
> - **Our work, on the other hand, is centered on when $\pi_\alpha$ attacks agent $\nu$, i.e., $\hat{\pi}\_\nu$, with a fixed $\hat{\pi}\_\alpha$.** Here, the adversarial training is geared towards equipping $\hat{\pi}\_\nu$ for these attacks.
>
> In other words, our paper together with [1] contributes to **different aspects** of this broader and challenging problem. It's valid to consider that the non-targeted agent could also be involved in adversarial training. Nonetheless, our approach remains relevant because:
> - Three-player games, even in zero-sum scenarios, are significantly harder than two-player games [2]. It's thus logical to first explore the two-player subproblem before delving into the more intricate three-player interactions. In essence, [1] investigates $(\hat{\pi}\_\alpha, \pi\_\alpha)$, and our study examines $(\hat{\pi}\_\nu, \pi\_\alpha)$. Each of these contributes to building a foundation for the three-player dynamics of $(\hat{\pi}\_\alpha, \pi\_\alpha, \hat{\pi}\_\nu)$.
> - Moreover, our defense algorithm holds even against unconstrained attackers, marking a notable advancement over existing research [3, 4, 5].
>
> Your question opens up an intriguing avenue for future research, which we're excited to explore. If you have any more thoughts or queries, we'd love to hear them. These discussions are invaluable in enriching our paper.
>
> ---
>
> [1] Tessler, Chen, Yonathan Efroni, and Shie Mannor. "Action robust reinforcement learning and applications in continuous control." International Conference on Machine Learning. PMLR, 2019.
>
> [2] Daskalakis, Constantinos, and Christos H. Papadimitriou. "Three-player games are hard." Electronic colloquium on computational complexity. Vol. 139. 2005.
>
> [3] Adam Gleave, Michael Dennis, Cody Wild, Neel Kant, Sergey Levine, and Stuart Russell. Adversarial policies: Attacking deep reinforcement learning. arXiv preprint arXiv:1905.10615, 2019.
>
> [4] Xian Wu, Wenbo Guo, Hua Wei, and Xinyu Xing. Adversarial policy training against deep reinforcement learning. In 30th {USENIX} Security Symposium ({USENIX} Security 21), 2021b.
>
> [5] Wenbo Guo, Xian Wu, Sui Huang, and Xinyu Xing. Adversarial policy learning in two-player competitive games. In International Conference on Machine Learning, pp. 3910–3919. PMLR, 2021.
>
> Paper2121 Authors

---

> > ### Comment · Reviewer_gybG · 2023-11-23
> >
> > I agree that the general version of the game might be out of the scope of the paper and appreciate the authors' clarification. I'll increase my score.

---

### Official Review · Reviewer_GBTd · 2023-10-31

**Soundness:** 3 good
**Presentation:** 3 good
**Contribution:** 3 good
**Rating:** 6
**Confidence:** 4

**Summary:**

This paper investigates adversarial attacks in multi-agent environments. Specifically, it considers a two-agent environment where one agent is controlled (hijacked) by an adversary, with the goal of minimizing the other agent's cumulative reward. The paper introduces the idea of a reduced capacity attack, where the adversary can only control the action of the hijacked adversary at each time step with a probability p. The paper derives a bound on policy discrepancy given p and investigates the stealthiness of an attack for a given p, where stealthiness is defined either as variation in the induced state distribution or as variation in the conditional transition dynamics. The paper then investigates adversarial training with timescale separation to achieve provably robust victim policies and derives a convergence bound that is polynomial with respect to the size of the state and action space. The paper then verifies the stealthiness of the proposed attack and the exploitability of robustified victim policies in two benchmark environments.

**Strengths:**

- the paper studies a very valid scenario for adversarial attacks in multi-agent systems
- the derived bounds on policy similarity, state distribution and conditional transition probabilities are insightful
- the conducted experiments indicate that their attacks are more stealthy than prior work
- the conducted experiments indicate that the robustification of victim policies results in significantly improved exploitability of victim agents

**Weaknesses:**

- Stealthiness is defined as the difference in state distributions. This is a step in the right direction and seems like a good proxy for real-world problems. However, two identical state distributions can come from very different trajectories (e.g. just inverting the order of states within all trajectories does not change the state distribution). This should be considered and discussed in more detail.
- The empirical results wrt stealthiness are very limited. Only three videos are provided for the qualitative comparison. A larger study and/or quantitative results would much improve the paper.
- The theoretical results are interesting, but the applicability of the given bounds to larger environments remains unclear.

**Questions:**

See weaknesses

---

> ### Author Response · Authors · 2023-11-20
> **Response to Reviewer GBTd**
>
> We thank Reviewer GBTd for the detailed comment and insightful feedback. We are encouraged the reviewer finds our paper "studies very valid scenario" and "derived bounds is insightful". We address Reviewer GBTd's concerns and questions below:
>
> ---
> ### [1/1] Response to weakness
>
> > Q1. Stealthiness is defined as the difference in state distributions. This is a step in the right direction and seems like a good proxy for real-world problems. However, two identical state distributions can come from very different trajectories (e.g. just inverting the order of states within all trajectories does not change the state distribution). This should be considered and discussed in more detail.
>
> - **Firstly, according to our standard definition of state occupancy, inverting the order of states within all trajectories could still change the state distribution.** The reason for this lies in Definition 2.1, where the probability is expressed as a sum of discounted probabilities. This formulation implies that states appearing later in a trajectory contribute less to the overall state distribution. Consequently, reversing the order of states within all trajectories can result in a subtle alteration of the state distribution. This distinction, rooted in the mathematical structure of the probability definition, highlights the impact of temporal positioning on state contributions.
> - **We agree with the reviewer that the discrepancy on trajectory distributions can be a more fine-grained metric**. The concept of a trajectory-level distribution is indeed capable of effectively characterizing the stealthiness of an adversarial policy. However, a significant reason for not adopting this metric in our analysis is the considerably larger size of the trajectory space compared to the state space. This poses a challenge in accurately computing discrepancies within trajectory distributions and in implementing constraints on them. The complexity and computational demands associated with managing this larger space are key factors in our decision to focus on state space metrics.
>
> ---
> > Q2. The empirical results wrt stealthiness are very limited. Only three videos are provided for the qualitative comparison. A larger study and/or quantitative results would much improve the paper.
>
> We thank the reviewer for the kind suggestions from the reviewer and shall add more videos in the link we provided in our paper.
>
> ---
> > Q3. The theoretical results are interesting, but the applicability of the given bounds to larger environments remains unclear.
> - **We believe for larger environments, our bounds still provide the correct intuitions.** For instance, in larger environments, the underlying principle that our attack budget effectively modulates the strength and stealthiness of the victim remains valid. Additionally, the implementation of asymmetric learning rates for the attacker and victim continues to enhance convergence. These concepts are consistently applicable, regardless of the environment's size.
> - **To scale to larger environments, the dependency on the state and actions space is usually replaced by the dimension of the state-action representations.** In the context of high-dimensional continuous environments, it is commonly assumed that state-action pairs can be represented within a $d$-dimensional space, as indicated by sources such as [1, 2]. Based on this premise, we posit that our theoretical findings can be extended to these settings. In these cases, the bounds of our results would depend solely on the dimensionality $d$, thereby offering a scalable and adaptable framework applicable to various high-dimensional environments.
> ---
> We greatly appreciate Reviewer GBTd's valuable feedback and constructive suggestions. We are happy to answer any further questions.
>
>
> [1] Jin, Chi, et al. "Provably efficient reinforcement learning with linear function approximation." Conference on Learning Theory. PMLR, 2020.
>
> [2] Agarwal, Alekh, et al. "Flambe: Structural complexity and representation learning of low rank mdps." Advances in neural information processing systems 33 (2020): 20095-20107.
>
> Paper2121 Authors

---

> > ### Comment · Reviewer_GBTd · 2023-11-21
> >
> > Thank you for the rebuttal, which addresses some of my concerns. I was trying to find the additional results regarding stealthiness, but wasnt able to. Can you please provide the link?

---

> ### Author Response · Authors · 2023-11-22
> **Response to Reviewer GBTd**
>
> We thank the reviewer for appreciating our responses. We apologize for not publishing our updated website correctly. The updated animations can be found in https://sites.google.com/view/stealthy-attack , where we have added two environments for further displaying the stealthiness of our approaches.
>
> If there are additional concerns from the reviewer, please do not hesitate to let us know. We believe your suggestion/comments can improve our paper greatly
>
> Paper2121 Authors

---

### Official Review · Reviewer_wz1v · 2023-11-01

**Soundness:** 2 fair
**Presentation:** 2 fair
**Contribution:** 3 good
**Rating:** 6
**Confidence:** 3

**Summary:**

This paper introduces a new attack formulation in multi-agent reinforcement learning, where an attacker can have partial control over agent  $\alpha$ to attack against a well-trained victim agent $v$. The paper highlights the limitations of existing attack models and proposes a more generalized attack framework that takes into account the level of attack budget and the attack model’s detectability. The paper offers a provably efficient defense with polynomial convergence to the most robust victim policy through adversarial training with timescale separation. The empirical results on two standard environment,: Kuhn Poker and RoboSumo, showcasing the proposed attack algorithm's effectiveness.

**Strengths:**

The proposed attack framework in multi-agent reinforcement learning takes into account the level of control the attacker has over the agent by introducing an "attack budget". This attack budget characterizes the partial control of the attacker and regulates the strength of the attacker in degrading the victim's performance. Regulating the attack budget allows us to manage the discrepancy between the state distributions and the transition dynamics before and after the attack, allowing for more stealthy attacks. The paper is well-written and presents the concepts, definitions, and analyses. A provably efficient defense with polynomial convergence have practical implications for the application of the proposed adversarial training.

**Weaknesses:**

In my understanding, the novelty of the paper is mainly the "partial control" and "attack budget", compared with the previous work: Gleave et al. (2019); Wu et al. (2021b); Guo et al. (2021). However, the partial control is previously introduced in PR-MDP, Tessler et al. (2019).

The motivation of the attack budget and partial control is not enough convincing. From the attack side, it is true that the setting of attack budget can improve the stealthiness, but it is hard to say that the proposed attack is optimal subject to the limitation of the discrepancy between the state distributions (or marginalized transition dynamics). From the side of the defense (adversarial training), there is no insight into the choice of the defense budget which is the budget used to train the robust victim. What is the benefit of introducing this defense budget? More discussion would be helpful.

The result in Figure 1 is confusing. 1. Why the bottom-right subfigure is empty? 2. Is original state distribution means that the distribution without any attack? If it is true, should not the original state distribution be same in the bottom and top subfigures? The original state distributions in top-mid and bottom-mid subfigures are different. Could you explain the reason? 3. Where do the results in Figure 1 come from? The paper only mentions that two attack polices are assessed under the same winning rate. However, according to the Figure 6, the unconstrained policy has higher wining rate.

**Questions:**

See in the weaknesses.

---

> ### Author Response · Authors · 2023-11-20
> **Response to Reviewer wz1v**
>
> We thank Reviewer wz1v for the detailed comment and insightful feedback. We are encouraged the reviewer finds our paper "well-written and presents the concepts, definitions, and analyses". We address Reviewer wz1v's concerns and questions below:
>
> ---
> ### [1/1] Response to weakness
> > Q1. In my understanding, the novelty of the paper is mainly the "partial control" and "attack budget", compared with the previous work: Gleave et al. (2019); Wu et al. (2021b); Guo et al. (2021). However, the partial control is previously introduced in PR-MDP, Tessler et al. (2019).
> - **Firstly, our contribution to the generalized attack formulation is not proposing "partial control", but understanding the impacts of a constrained formulation on stealthiness and defenses in a multi-agent setting.** The fact that PR-MDP also considered the partial control does not weaken our novelty but well justifies our problem formulation since our discussions on how a constrained formulation could bring benefits to stealthiness and defenses is unique to our settings of adversarial policies, which are not discussed in PR-MDP, Tessler et al. (2019).
> - **Our provable defense algorithm is one of our key novelties, which does not appear in Gleave et al. (2019); Wu et al. (2021b); Guo et al. (2021); Tessler et al. (2019).** Note that Gleave et al. (2019); Wu et al. (2021b); Guo et al. (2021) only considered heuristic defense mechanism while the defense of Tessler et al. (2019) highly depends on the single-agent setup such that the policy iteration algorithm can be efficient. Therefore, our paper improves the defense in Gleave et al. (2019); Wu et al. (2021b); Guo et al. (2021) and the algorithm is significantly different from that of Tessler et al. (2019), thus highlighting the novelty of our paper.
> ---
> > Q2.1. The motivation of the attack budget and partial control is not enough convincing. From the attack side, it is true that the setting of attack budget can improve the stealthiness, but it is hard to say that the proposed attack is optimal subject to the limitation of the discrepancy between the state distributions (or marginalized transition dynamics).
>
> - **We believe the discrepancy between the state distributions (or marginalized transition dynamics) itself is a good metric to evaluate stealthiness but not a natural metric to model real-world constraints that attackers could encounter.**  As in our response to Q1, our paper focuses on the understanding of a generalized attack formulation for real-world applications that often exhibit certain natural constraints on the attacker, instead of proposing a specific type of attacking strategy to ensure optimal stealthiness.
> - **The optimal adversarial policy subject to the stealthiness conditions suggested by the reviewer could potentially violate the common and well-accepted constraint on the attacker (Equation 3.2 and also Tessler et al. (2019)).** Therefore, we believe developing such optimal attacks under the stealthiness constraints is a promising and interesting direction but beyond the scope of our paper of understanding the generalized formulation and provable defense mechanisms of adversarial policies.
>
> > Q2.2. From the side of the defense (adversarial training), there is no insight into the choice of the defense budget which is the budget used to train the robust victim. What is the benefit of introducing this defense budget? More discussion would be helpful
>
> **According to our experiments, when the attack budget and defense budget are matched, the victim exhibits the strongest robustness.** We have provided experimental evidence in **Table 4** that when the defense budget matches the attack budget, the victim achieves the best performance. In other words, if we do not introduce the defense budget and just consider an unconstrained one in adversarial training, the performance of the victim can be suboptimal when the real test-time attackers are highly constrained. **In summary, introducing a defense budget gives the flexibility to defend against adversarial policies of different attack levels**.
>
> ---

---

> ### Author Response · Authors · 2023-11-20
> **Response to Reviewer wz1v**
>
> > Q3.1. The result in Figure 1 is confusing. 1. Why the bottom-right subfigure is empty?
>
> From the authors' side, we are not aware of any empty figures and we suggest the reviewer try to refresh/reopen our pdf. In case the bottom-right subfigure is still empty, we also provide Figure 1. in the anonymous link (https://drive.google.com/file/d/1LCo824MlmxMT4bOe6QaxNrMAEn3buY9L/view?usp=sharing).
>
> > Q3.2 Is original state distribution means that the distribution without any attack? If it is true, should not the original state distribution be same in the bottom and top subfigures? The original state distributions in top-mid and bottom-mid subfigures are different. Could you explain the reason?
>
> Firstly, what the reviewer understands is correct that the original state distribution means there is no attack.
>
> Secondly, the original state distribution is **marked by the blue points**. Therefore, the original state distributions in top-mid and bottom-mid subfigures are exactly the same and we are not aware of any differences from Figure 1. We guess the reason why the reviewer feels they are different may be because some blue points in the top-mid subfigure overlap with the orange points.
>
> > Q3.3. Where do the results in Figure 1 come from? The paper only mentions that two attack polices are assessed under the same winning rate. However, according to the Figure 6, the unconstrained policy has higher wining rate.
> - **Firstly, we don't claim that varying attack budgets will yield the same rewards or win rates.** What the reviewer understands is correct. Our Appendix E4 shows the trade-off: reduced budgets boost stealth but might limit the potential of achieving higher rewards.
> - **Secondly, the selected policies for the unconstrained attack do not necessarily reflect the algorithms' final state**. It's important to clarify that the policies selected for comparison in the unconstrained attack scenario do not necessarily represent the ultimate outcome of the algorithms. For instance, if an adversarial policy developed through our method attains a win rate of 0.7, we juxtapose it with an unconstrained adversarial policy also having a win rate of 0.7. However, such an unconstrained policy may not reflect the final convergence point of the baseline algorithm. The key takeaway is that when an attacker prioritizes stealthiness while targeting a specific win rate, our constrained attack methodology demonstrates superior stealth performance.
> .
>
> ---
> We greatly appreciate Reviewer wz1v's valuable feedback and constructive suggestions. We are happy to answer any further questions.
>
> Paper2121 Authors

---

> > ### Comment · Reviewer_wz1v · 2023-11-22
> >
> > Thank you for the detailed response. Most of my concerns are addressed. I increased my score to reflect that.

---

> ### Author Response · Authors · 2023-11-22
> **Thanks for your reply!**
>
> We are happy that your concerns are addressed. We believe your dedicated efforts have greatly improved our paper!
>
> Paper2121 Authors

---

### Official Review · Reviewer_RFDQ · 2023-11-03

**Soundness:** 3 good
**Presentation:** 3 good
**Contribution:** 3 good
**Rating:** 8
**Confidence:** 3

**Summary:**

The paper studies RL in an adversarial setting. In particular, the focus is on multi- (specifically 2) agent setting in which an agent $\alpha$ has the goal of harming the other agent $\nu$. So, the goal of $\nu$ is to find the most return (based on the discount factor $\lambda$) while $\alpha$ aims to the opposite direction for $\nu$.

The paper then models a form of "budget" for the adversarial agent, but letting them impose a policy that is a probabilistic (linear) combination of the original (non-adversarial) policy and a fully malicious adversarial one. So, a parameter $\epsilon \in [0,1]$ is introduced here to control this budget.

Some initial (proposition) results are proved showing that with small $\epsilon$, the effect of adversary's strategy is limited (which is expected, but proving them formally is not completely trivial).

The paper then aims at designing a form of adversarial training. They propose a method with "two separated timescale" meaning that the optimization is done in iterations between two types of operations. In one type of operation one fixes the adversary's strategy and optimizes the victim policy, and then switches to to the opposite.

The paper proves converges properties for their defense, and then move to do experiments to show the capability of the defense in different senses (such as convergence and robustness of the final policy).

**Strengths:**

Studying RL's robustness in the multi-agent setting is a natural in interesting problem. The formulation is clean and the proposed defense is analyzed well, both theoretically and experimentally.

**Weaknesses:**

Despite numerous references to previous work, there is no clean depicted picture of how this work on multi agent robustness fits in the context of previous work. I am not complaining for lack of references here, I just think the presentation should be more clear.

Some corrections needed (see comments below as well).

some example of typos:
Def 5.1 : "has" -> 'have"

paragraph "(c) Connection to action adversarial RL." in page 4, has "\tilde{\pi}\alpha" in which alpha needs to be the index.

Definition 5.2. : please give an intuition of what this is, before giving a full line math formula.

**Questions:**

You say "performance. This contrasts with supervised learning attacks, where small perturbations can result in large performance shifts." but this seems to be only applicable to test time attacks, as poisoning has a limited effect if it is limited to eps-fraction of data (e.g, ERM will be effected by eps increase in its probability of error as well).

When you say "However, it has been demonstrated that while re-training against a specific adversarial policy does augment robustness against it, the performance against other policies may be compromised." what is the reference?

---

> ### Author Response · Authors · 2023-11-20
> **Response to Reviewer RFDQ**
>
> We thank Reviewer RFDQ for the detailed comment and insightful feedback. We are encouraged the reviewer finds our paper "The formulation is clean and the proposed defense is analyzed well, both theoretically and experimentally". We address Reviewer RFDQ's concerns and questions below:
>
> ---
> ### [1/2] Response to weakness
> > Q1. Despite numerous references to previous work, there is no clean depicted picture of how this work on multi agent robustness fits in the context of previous work. I am not complaining for lack of references here, I just think the presentation should be more clear.
>
> We thank for the kind suggestions of the reviewers. The most closely related works to us are [1, 2, 3] and we discuss how our paper fits in the context of them as follows:
>
> - **Our research offers a more comprehensive attack formulation compared to references [1, 2, 3], incorporating additional aspects such as stealth**. Our paper advances the understanding of adversarial policies in practical scenarios, where attackers are often limited by physical world constraints and motivated to maintain stealth.
> - **We introduce the first provably effective defense algorithm against adversarial policy threats**, a significant step beyond the primarily heuristic approaches of prior work, which mainly focus on retraining against specific victims.
> ---
> > Q2. Some corrections needed (see comments below as well). some example of typos: Def 5.1 : "has" -> 'have" paragraph "( c ) Connection to action adversarial RL." in page 4, has "\tilde{\pi}\alpha" in which alpha needs to be the index.
>
> We thank the reviewer for pointing out these typos and we revised our paper accordingly.
>
> ---
> > Q3. Definition 5.2. : please give an intuition of what this is, before giving a full line math formula.
>
> We apologize the brevity in our explanations of these concepts. Definition 5.2 essentially provides an intuitive measure of the difficulty an agent faces while exploring the environment. This is achieved by comparing the stationary state occupancy frequencies under certain policies against the initial state distribution. In simpler terms, a smaller value of this quantity indicates that the environment is more easily explorable. We have also included more detailed explanations in our revised paper.
> ### [2/2] Response to questions
>
> > Q4. You say "performance. This contrasts with supervised learning attacks, where small perturbations can result in large performance shifts." but this seems to be only applicable to test time attacks, as poisoning has a limited effect if it is limited to eps-fraction of data (e.g, ERM will be effected by eps increase in its probability of error as well).
>
> We agree with the reviewer that our argument mainly applies to test-time attacks. We have made our argument more accurate in the revision. Since our setting is focusing on test-time attacks for RL, we are comparing it with test-time attacks of supervised learning. For training-time attacks, the effect of the poisons can have very different effects on the victim policy, which we believe is an interesting future work.
>
> ---
> > Q5. When you say "However, it has been demonstrated that while re-training against a specific adversarial policy does augment robustness against it, the performance against other policies may be compromised." what is the reference?
>
> This has been validated experimentally by [1] in Section 6. Intuitively, if the victim is retrained against a specific attacker, its policy might be overfitted to the trained attacker. Hence, this highlights the necessity of more advanced adversarial training algorithms instead of naive retraining.
>
> ---
> We greatly appreciate Reviewer RFDQ's valuable feedback and constructive suggestions. We are happy to answer any further questions.
>
>
> [1] Adam Gleave, Michael Dennis, Cody Wild, Neel Kant, Sergey Levine, and Stuart Russell. Adver- sarial policies: Attacking deep reinforcement learning. arXiv preprint arXiv:1905.10615, 2019.
>
> [2] Xian Wu, Wenbo Guo, Hua Wei, and Xinyu Xing. Adversarial policy training against deep reinforcement learning. In 30th {USENIX} Security Symposium ({USENIX} Security 21), 2021b.
>
> [3] Wenbo Guo, Xian Wu, Sui Huang, and Xinyu Xing. Adversarial policy learning in two-player competitive games. In International Conference on Machine Learning, pp. 3910–3919. PMLR, 2021.
>
> Paper2121 Authors

---

> ### Comment · Reviewer_RFDQ · 2023-11-22
> **Ack**
>
> Thanks for the responses. I find them helpful. Please add the relevant discussions (from your response) to the paper.
> (I increased my score.)

---

> ### Author Response · Authors · 2023-11-22
> **Thanks for your reply!**
>
> We are happy to hear that our responses are helpful! We shall revise our paper accordingly and believe your dedicated efforts have greatly improved our paper!
>
> Paper2121 Authors

---

### Meta-Review · Area_Chair_yfMh · 2023-12-03

**Metareview:**

This paper studies adversarial attacks and defenses on RL agents. In contrast to prior works that focus on attacks that perturb an agent's perceived state, action, and rewards, this paper investigate the angle of attack where the adversary controls an another agent which in turns affects the environment the victim agent lives in. Another novelty is the explicit characterization of stealthiness of the attack, which is often an important angle for defense in real applications.

Overall, this is a solid paper worthy of publication.

**Justification For Why Not Higher Score:**

Neither the multi-agent perspective of attack nor the involve of attack budget is new in the literature of adversarial robustness RL, therefore the novelty is only marginal.

**Justification For Why Not Lower Score:**

NA

---

### Decision · Program_Chairs · 2024-01-16

Accept (poster)